

# Quantitative mapping and predictive modelling of Mn-nodules' distribution from hydroacoustic and optical AUV data linked by Random Forests machine learning.

Iason-Zois Gazis[1], Timm Schoening[1], Evangelos Alevizos[1], Jens Greinert[1,2]

[1]GEOMAR Helmholtz Centre for Ocean Research Kiel, Wischhofstrasse 1-3, 24148 Kiel, Germany
[2]Christian-Albrechts University Kiel, Institute of Geosciences, Ludewig-Meyn-Str. 10-12, 24098 Kiel, Germany

*Correspondence to*: Iason-Zois Gazis (igazis@geomar.de)

**Abstract.** In this study, high-resolution bathymetric multibeam and optical image data, both obtained within the Belgian manganese (Mn) nodule mining license area by the autonomous underwater vehicle (AUV) Abyss, were combined in order to create a predictive Random Forests (RF) machine learning model. AUV bathymetry reveals small-scale terrain variations, allowing slope estimations and calculation of bathymetric derivatives such as slope, curvature, and ruggedness. Optical AUV

imagery provides quantitative information regarding the distribution (number and median size) of Mn-nodules. Within the area considered in this study, Mn-nodules show a heterogeneous and spatially clustered pattern and their number per square meter is negatively correlated with their median size. A prediction of the number of Mn-nodules was achieved by combining information derived from the acoustic and optical data using a RF model. This model was tuned by examining the influence of the training set size, the number of growing trees (*ntree*) and the number of predictor variables to be randomly selected at

each RF node (*mtry*) on the RF prediction accuracy. The use of larger training data sets with higher *ntree* and *mtry* values increases the accuracy. To estimate the Mn-nodule abundance, these predictions were linked to ground truth data acquired by box coring. Linking optical and hydro-acoustic data revealed a non-linear relationship between the Mn-nodule distribution and topographic characteristics. This highlights the importance of a detailed terrain reconstruction for a predictive modelling of Mn-nodule abundance. In addition, this study underlines the necessity of a sufficient spatial distribution of the optical data

to provide reliable modelling input for the RF.



# 1. Introduction

High-resolution quantitative predictive mapping of the distribution and abundance of manganese nodules (Mn-nodules) is of
interest for both the deep-sea mining industry and scientific fields as marine geology, geochemistry, and ecology. The
distribution and abundance of Mn-nodules are affected by several factors such as local bathymetry (Craig 1979; Kodagali,
1988; Kodagali and Sudhakarand, 1993; Sharma and Kodagali, 1993), sedimentation rate (Glasby, 1976; Frazer and Fisk,
1981; von Stackelberg and Beiersdorf 1991; Skornyakova and Murdmaa, 1992), availability of nucleus material (Glasby,
1973), and bottom current strength (Frazer and Fisk, 1981; Skornyakova and Murdmaa, 1992). As a consequence, the
distribution and abundance of Mn-nodules is heterogeneous (Craig, 1979; Frazer and Fisk, 1981; Kodagali, 1988; Kodagali
and Sudhakar, 1993; Kodagali and Chakraborty, 1999; Kuhn et al., 2011), even on fine scales of 10 to 1,000 m (Peukert et
al., 2018a; Alevizos et al., 2018). This increases the difficulty for quantitative predictive mapping using remote sensing
methods. Vast areas of the sea bottom can be mapped by ship-mounted, multibeam echo-sounder systems (MBES). State-of-
the-art MBES systems feature a low frequency (12 kHz) and can map ca. 300 km$^2$ of seafloor in 4,500 m water depth per
hour. Hence, low-resolution regional maps can be created at a grid cell size of 50 to 100 m within which the main Mn-nodule
occurrence can become apparent (Kuhn et al., 2011; Rühlemann et. al., 2011; Jung et al., 2001). A general separation in
areas of high and low abundance (kg/m²) of Mn-nodules seems possible, especially in flat areas where sedimentological
changes and physical influences on the footprint size and incidence angle of the transmitted acoustic pressure wave can be
corrected accurately (De Moustier, 1986; Kodagali and Chakraborty, 1999; Chakraborty and Kodagali, 2004; Kuhn et al.al.,
2010 and 2011, Rühlemann et al., 2011 and 2013). However, the patchy distribution of Mn-nodules in size and number at
meter-scale cannot be resolved with ship-mounted MBES data (Petersen, 2017). For an operational resource assessment, a
higher resolution of few meters grid cell size is needed to supply accurate depth, slope, and Mn-nodule distribution
variability (Kuhn et al., 2011). Supplementary to the spatial mapping by acoustic sensors, point-based measurements from
box-corer samples are used as ground truth data for training and validation of geostatistical techniques (e.g. kriging) in order
to create quantitative maps of Mn-nodule abundance (Mucha et al., 2013; Rahn, 2017). However, the generally low number
of ground-truth samples during surveys (usually below 10), their limited sampling area (typically 0.25m$^2$) and the relatively
large distance between them (> 1nmi) prevent an accurate correlation with the ship-based MBES data and thus a good
prediction of the total Mn-nodules' mass and distribution in large areas (Petersen, 2017). Importantly, the sparse sampling
with box corers affects the performance of interpolation and geostatistical techniques, which are typically applied during
data analysis (Li and Heap, 2011 & 2014; Kuhn et al., 2016). In this article, we address this challenge by combining high-
resolution hydroacoustic and optical data acquired with an Autonomous Underwater Vehicle (AUV) and connecting those
data with a Machine Learning (ML) algorithm (here Random Forests), in order to predict the spatial distribution of the
number of Mn-nodules per square meter. Unlike geostatistical methods, ML can be used to incorporate information from
different bathymetric derivative layers and to detect complex relationships among predictor variables without making any
prior assumptions about the type of their relationship or value distribution (Garzón et al., 2006; Lary et al., 2016). First





predictions have already been achieved by Knobloch et al. (2017), Vishnu et al., 2017 and Alevizos et al., 2018. Here, we present a complete data analysis workflow for potential mining operations (Figure 1).

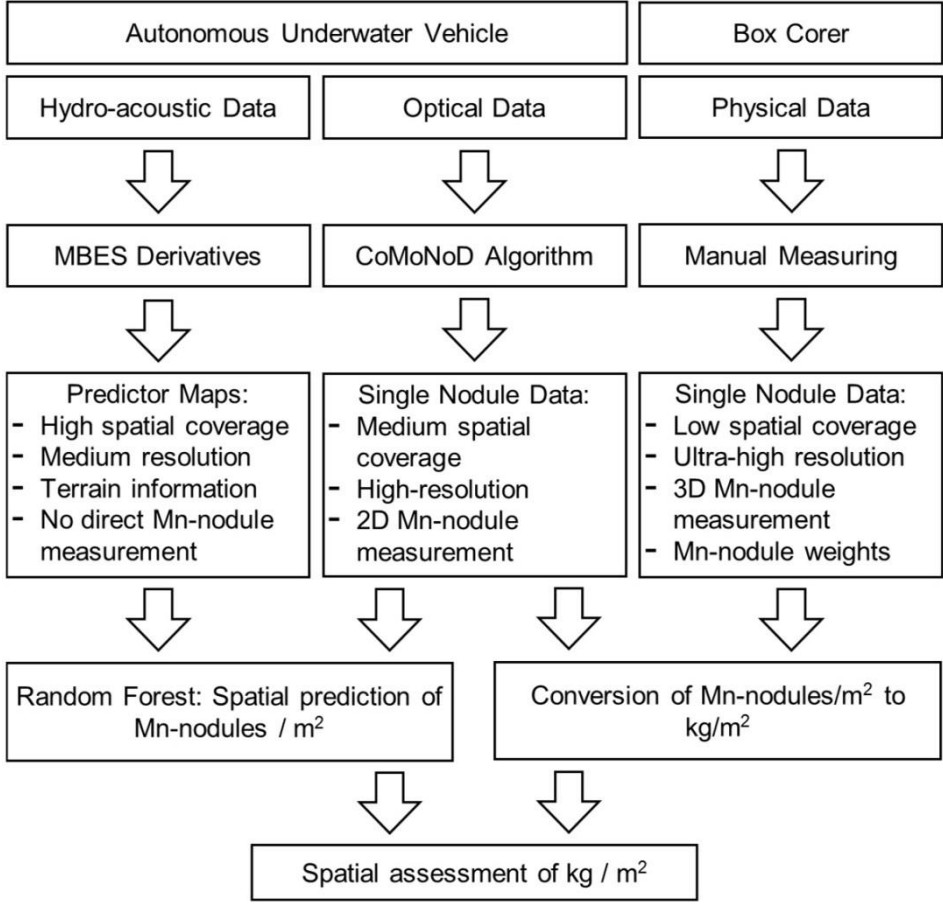

**Figure 1**. Schematic workflow of the data sets used in this study to enable the spatial assessment of Mn-nodules inside the study area. The medium resolution of AUV MBES (m scale) is referring to the comparison of the optical and physical data (cm scale).

## 1.1 AUV hydroacoustic mapping

AUVs have proven their usefulness for multibeam data acquisition in the deep-sea environment (Grasmueck et al., 2006; Deschamps et al., 2007; Haase et al., 2009; Wynn et al., 2014; Clague et al., 2014 and 2018; Pierdomenico et al., 2015; Peukert et al., 2018a). They achieve higher spatial and vertical resolution compared to ship-mounted MBESs. This is due to their operation close to the seafloor which results in a smaller footprint at a given beam angle and enables the use of higher frequencies (Henthorn et al., 2006; Mayer, 2006; Caress et al., 2008; Paduan et al., 2009). Additionally, AUVs avoid



problems like near-surface turbulences, bubbles, ship-noise and strong sound velocity changes (Kleinrock et al., 1992a and
1992b; Jakobson et al., 2016; Paul et al., 2016). They work independently from the surface vessel and operate at a stable
altitude. AUVs can efficiently conduct a dive pattern of dense survey lines and thus reduce survey effort and costs (Chance
et al., 2000; Bellingham, 2001; Bingham et al., 2002; Danson, 2003; Roman and Mather, 2010). High-resolution bathymetry
enables computing bathymetric derivatives like slope and rugosity with a similarly high resolution. These derivatives play an
important role in predicting Mn-nodules' distribution and abundance (Craig, 1979; Kodagali, 1988; Skornyakova and
Murdmaa, 1992; Kodagali and Sudhakar, 1993, Sharma & Kodagali, 1993; Ko et al., 2006). Until now, few studies have
examined the correlations between Mn-nodules and bathymetry, its derivatives, and backscatter response in AUV scale
(Okazaki and Tsune, 2013; Peukert et al., 2018a; Alevizos et al., 2018) promoting such data as key for predictive modelling
of Mn-Nodule occurrence (e.g. Alevizos et al., 2018).

## 1.2 Underwater optical data

Underwater optical data have generally played an important role in the qualitative analysis of the seafloor features and for
the specific task of assessing Mn-nodules' distribution explicitly (Glasby, 1973; Rogers, 1987; Skornyakova and Murdmaa,
1992; Sharma et al., 1993). The development of automated detection algorithms enabled quantitative optical image data
analysis and subsequent statistical interpretation of Mn-nodule densities. The spatial coverage of optical imaging is much
higher than for box core sampling. The data resolution remains high enough to reveal the high variance in the spatial
distribution of nodules at meter scale. Thus optical data can fill the investigation gap between ground truth sampling and
hydro-acoustic remote sensing (Sharma et al., 2010 and 2013; Schoening et al., 2012a, 2014, 2015, 2016 and 2017a; Kuhn
and Rathke, 2017). Moreover, mosaicking of optical data could reveal mining obstacles such as outcropping basements or
volcanic pillow lava flows. In addition, seafloor photos are the source for evaluating benthic fauna occurrences and related
habitats on a wider area (Schoening et al., 2012b; Durden et al., 2016).

## 95    1.3 Box corer sampling

Box coring is common to obtain physical samples of Mn-nodules and sediments for resource assessments and biological
studies. While optical data reveal only the exposed and semi-buried Mn-nodules, box corers collect the top 30-50 cm of the
seafloor with minimum disturbance, allowing an accurate measure of the Mn-nodules' abundance ($kg/m^2$). Box coring data
are used for training and validation in geostatistical methods for quantitative and spatial predictions of Mn-nodules (e.g.
Mucha et al., 2013; Knobloch et al., 2017). The representativeness of box coring data is disputable as few deployments can
be made due to time constraints (ca. 4h per core) and as the spatial coverage of one sample is rather low (ca. $0.25m^2$).



### 1.4 Random Forests

Random Forests (RF) is an ensemble machine learning (ML) method composed of multiple weaker learners, namely
classification or regression trees (Breiman, 2001a). Within RF an ensemble of distinct tree models is trained using a random
subsample of the training data for each tree until a maximum tree size is reached. In each tree, each node is split using the
best among a subset of predictors randomly chosen at that node instead of using the best split among all variables (Liaw &
Wiener, 2002). Thus, the process is double-randomized which further reduces the correlation between trees. About two
thirds of the training data are used to tune the RF while the remaining 'out-of-bag' (OOB) samples are used for an internal
validation. By aggregating the predictions of all trees (majority votes for classification, the average for regression) new
values can be predicted. This aggregation keeps the bias low while it reduces the variance, resulting in a more powerful and
accurate model. RFs have the ability to estimate the importance of each predictor variable which enables data mining of the
high-dimensional prediction data. Terrestrial studies use RFs in prospectivity mapping of mineral deposits (Carranza and
Laborte, 2015a; 2015b; 2016; Rodriguez-Galiano et al., 2014 and 2015). In the marine environment RFs have been used to
combine MBES bathymetry, backscatter, their derivatives, sediment sampling, and optical data for various seabed
classification and regression tasks (e.g. Li et al., 2010; Li et al., 2011a; Che Hasan et al., 2014; Huang et al., 2014). Further
studies showed the robustness of RFs for selected data sets compared to other ML algorithms (Che Hasan et al., 2012;
Stephens and Diesing, 2014; Diesing and Stephens, 2015; Herkul et al. 2017), as well as to geostatistical and deterministic
interpolation methods (Li et al., 2010, 2011b and 2011b; Diesing et al., 2014).

## 2. Study Area

The study area lies in the Clarion–Clipperton Zone (CCZ; ca. $4x10^6$ km$^2$) in the Eastern Central Pacific Ocean. The CCZ
triggered scientific and industrial interest for several decades due to its high resource potential in Mn-nodules deposits (Hein
et al., 2013; Petersen et. al., 2016) with an average nodule abundance of 15 kg/m$^2$ (SPC, 2013). At the time of writing, the
International Seabed Authority (ISA) has granted 17 exploration licences inside the CCZ (Figure 2a). The study area
described here is part of the Belgian GSR license area (Figure 2b) and will be referred to as Block G77 (Figure 2c). Overall,
this part of the Belgian license area has high bathymetric range, and complex morphology, due to the presence of submarine
volcanoes, solitary seamounts and seamount chains. Block G77 is characterized by a low bathymetric range (77 m) and
mostly gentle slopes (95% of the area below 5°). An exception is located in the eastern part, where sub-recent small-scale
volcanic activity created 15 cone-shaped morphological features of up to 55m height and 150m width that are clustered in an
area of circa 700 m x 380 m.  Despite the gentle slopes, bock G77 is characterized by an uneven micro-relief (according to
Dikau (1990) scale) especially in the western part, where small (2-4 m) depressions coexist next to short (2-4 m) protrusions.
In the central part, a 30m high elevation acts as a natural barrier between the western part of the study area that features more
relief and the eastern part that is deeper has less relief (Figure 2c).





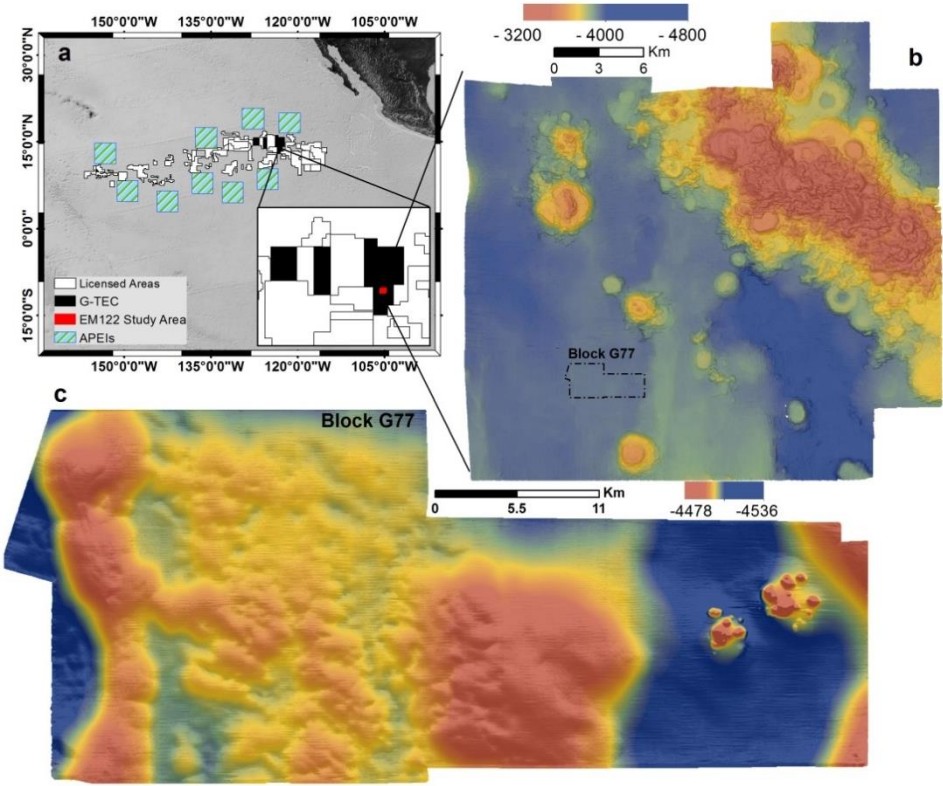

**Figure 2.** a) Areas of Particular Environmental Interest (APEIs), licensed areas (white) and the Belgium / GSR licenses area (black) within the CCZ. b) Regional bathymetric map of the study area, created by the EM 122 MBES on R/V SONNE (cruise SO239). c) Block G77, mapped by AUV Abyss with a Teledyne Reson Seabat 7125 MBES.

## 3. Methodology

### 3.1 Hydroacoustic Data Acquisition & Post Processing

The data (Greinert, 2016) were collected in March 2015 during cruise SO239 EcoResponse (Martínez Arbizu & Haeckel, 2015) with the German Research Vessel Sonne. Ship-based mapping was conducted with a hull-mounted Kongsberg EM 122 MBES (12 kHz, 0.5° along- and 1° across-track beam angle, 432 beams with 120° swath angle). High-resolution MBES data were acquired with AUV Abyss (GEOMAR, 2016) inside Block G77 equipped with a Teledyne Reson Seabat 7125 MBES (200 kHz, 2° along- and 1° across-track beam angle, 256 beams with 130° swath angle). The data (60 km of survey

lines) were acquired from 50m altitude and with 100% swath overlap resulting in an insonification of 9.5km$^2$. Post-processing of the AUV data was conducted with the Teledyne PDS2000 software for data conversion of the raw data into s7k and GSF format. Further multibeam processing (sound velocity calibration, pitch/roll/yaw/latency artifacts correction) was performed using the Qimera™ software. The largest uncertainties during AUV operations result from inaccurate



navigation and localization in the deep-sea environment (Paull et al., 2014). AUV Abyss has a combination of five different
systems for navigation and positioning: Global Positioning System (GPS) when at the sea surface, Doppler Velocity Log
(DVL) when 100 m or less from the ground, Inertial Navigation System (INS), Long Baseline Acoustic Navigation (LBL)
and dead reckoning (GEOMAR, 2016). Each system has its own limitations that contribute to the total navigation error
(Sibenac et al., 2004; Chen et al., 2013) that generally results in positioning drifts over time. Consequently, this affects the
position accuracy of the MBES and photo data. Our AUV MBES depth data processing guarantees both the removal of those
artifacts and the fitting of adjacent tracks and absolute geo-referencing by the correlation of ship-based and AUV-based
bathymetric data. However, the backscatter data were excluded from the modelling procedure due to further artifacts and the
overall poor quality. The output grid cell size for the analyses was set to 3m x 3m. The depth raster was exported as ASCII
format for further analysis in SAGA GIS v.6.3.0. SAGA includes numerous tools that focus on DEM and Terrain Analysis
(Conrad, 2015). Eight bathymetric derivatives were computed (Table 1) with the SAGA algorithms (Appendix A).

**Table 1.** The bathymetric derivatives computed in SAGA GIS and used as predictor variables.

| Derivative | Description |
|---|---|
| Slope (S) | The first derivative of the bathymetry and describes the steepness of a surface. |
| Plan Curvature (Pl.C) | The second derivative of the bathymetry and perpendicular to the direction of the maximum slope (Zevenbergen and Thorne, 1987). |
| Profile Curvature (Pr.C) | The second derivative of the bathymetry and parallel to the direction of the maximum slope (Zevenbergen and Thorne, 1987). |
| Topographic Position Index (TPI) | Compares the elevation of a single pixel to the average of multiple cells surrounding it in a defined distance (Weiss, 2001). |
| Broad-scale (TPI_B) | Distance: 150-400m |
| Medium-scale (TPI_M) | Distance: 50-150m |
| Fine-scale (TPI_F) | Distance: 0-50m |
| Concavity (C) | In each cell its value is defined as the percentage of concave downward cells within a constant radius (Iwahashi & Pike, 2007). Here, a 10 cell radius was used. |
| Terrain Ruggedness Index (TRI) | A quantitative measure of surface heterogeneity and can be explained as the sum change in elevation between a central pixel and its neighborhood (Riley et al, 1999). Here, a 10 cell radius was used. |

**3.2 Optical Data Acquisition & Post Processing**

High-resolution optical data (20.2 Megapixels) was acquired by the DeepSurveyCamera system on board AUV Abyss
(Kwasnitschka et al., 2016). During image acquisition the altitude above ground was 5 to 11 m, resulting in an overlap
between the images of ca. 60% in each direction. In total, 11,276 photos acquired in block G77 (Greinert, 2017) and



analysed with the automated image analysis algorithm CoMoNoD (Schoening et al., 2017a, 2017b and 2017c). For each image this algorithm delineates each individual Mn-nodule and provides quantitative information on each nodule (size in $cm^2$, alignment of main axis, geographical coordinate of the nodule). This information is further aggregated per image to

provide the average number of Mn-nodules per square meter (Mn-nodules/$m^2$), the nodule coverage of the seafloor in percent and the nodule size distribution in $cm^2$ size quantiles. The algorithm has successfully been applied for quantitative assessment and predictive modelling of Mn-nodules (Peukert et al., 2018a, Alevizos et al., 2018). Nevertheless, the derived number of Mn-nodules/$m^2$ is subject to uncertainties due to the limitations of the CoMoNoD algorithm and the non-constant altitude of the AUV, especially in areas with slopes. The CoMoNoD algorithm cannot detect sediment-covered Mn-nodules

due to the low or non-existent contrast. It may count two or more adjacent small Mn-nodules as one big nodule or misinterpret benthic fauna or rock fragments with similar visual features as Mn-nodules. The CoMoNoD algorithm fits an ellipsoid around each detected Mn-nodule, which limits the first two disadvantages as it splits huge Mn-nodules and accounts for potentially buried parts (see discussions in Schoening et al., 2017a). In general, the first two disadvantages lead to underestimations while the third one results in an overestimation of the number of Mn-nodules per $m^2$. These limitations

are common and the need for corrections between optical and box-corer data has been acknowledged (Sharma and Kodagali, 1993; Sharma et al., 2010 and 2013; Tsune and Okazaki, 2014; Kuhn and Rathke, 2017). Recent studies show that the difference between image estimates and the abundance in box corer data (due to sediment covered Mn-nodules) can be two to four times higher (Kuhn and Rathke, 2017). In this study, none of the box-corers was obtained exactly at a location for which optical data exists, thus no direct comparison and verification exist. Taking box corer samples for verification requires

Ultra Short Baseline (USBL) navigation and imaging of the seafloor prior to the physical sampling. The effects of the non-constant flying altitude on the detection of Mn-nodules per $m^2$ are explained in detail below. For each photo location, the depth and the bathymetric derivative values were extracted from the hydro-acoustic data. As no absolute geo-referencing could be performed for the AUV-based photo surveys, drifting sensor data will have an effect on the alignment between bathymetric and photo information, which was considered wile interpreting the results.

### 3.3 Data Exploration and Spatial Analysis

The data exploration, spatial plotting and analysis was performed with ArcMap™ 10.1, PAST v3.19 (Hammer et al., 2001), and R (R, 2008). All data were projected as UTM Zone 10N coordinate system (to enable spatial analysis). The existence of spatial autocorrelation in the distribution of Mn-nodules/$m^2$ was examined by the Global Moran's Index (GMI) and Anselin Local Moran's Index (LMI). Both, GMI (Moran, 1948 and 1950) and LMI (Anselin, 1995) are well-established for

examining the overall (global) and local spatial autocorrelation, respectively (e.g. Goodchild, 1986; Fu et al., 2014). GMI attains values between -1 and 1 with high positive values indicating strong spatial autocorrelation. High positive LMI index values indicate a local cluster. This cluster could be a group of observations with high-high (H-H) or low-low (L-L) values regarding the examined variable. A high negative index value implies local outliers, like high-low (H-L) or low-high (L-H) clusters, in which an observation has a higher or lower value in comparison to its adjacent observations. Both Moran's Index



analyses were performed in ArcMap™ 10.1 (for parameter settings see Appendix A). One decimal was retained in the presentation of the results from statistical analysis and RF modelling.

### 3.4 Box corer Data

A total of five box-corers (0.5m x 0.5m surface area) were obtained close to the study area (coordinates not given due to confidentiality). However, one is located within Block G77 (Figure 3a); this is the result of independent sampling schemes and purposes during the cruise. Nevertheless, all box core samples (maximum distance <1.5km), were analyzed and used for further analyses. In each box-corer, the number, size, and weight of nodules were measured and the abundance ($kg/m^2$) was estimated (mean value: 26.5kg/m²). The total number of Mn-nodules within each box corer was compared with the number of Mn-nodules on the surface resulting in an average ratio of 1.32 (Table 2). This means that ≈ 25% of the nodules are not seen on the surface but are completely buried within the sediment (down to a depth of about 15cm).

**Table 2.** The number of Mn-nodules on the sediment surface, the total number of Mn-nodules per box core, the ratio of those two values, and the distance of the box corer deployments from the study area in block G77.

| box corer station | total number of Mn-nodules | number of Mn-nodules at the surface | ratio | abundance ($kg/m^2$) | distance from G77 area (km) |
|---|---|---|---|---|---|
| BC20 | 40 | 27 | 1.5 | - | 0 |
| BC21 | 67 | 58 | 1.1 | 27.1 | 1.4 |
| BC22 | 29 | 21 | 1.4 | 27.1 | 0.6 |
| BC23 | 32 | 20 | 1.6 | 25.2 | 0.1 |
| BC24 | 17 | 16 | 1.0 | - | 1 |
| **Average** | **37** | **28** | **1.32** | **26.5** | **0.6** |

### 3.5 RF Predictive modelling

The RF modelling was performed with the Marine Geospatial Ecology Tools (MGET) toolbox in ArcMap™ 10.1. MGET (Roberts et al., 2010) uses the *randomForests* R package for classification and regression (Liaw and Wiener, 2002). Our target variable (number of Mn-nodules/m²) is continuous, so regression was applied. We followed the three main steps to establish a good model by selecting predictor variables, calibration/training of the model and finally validating the model results.

Selection of Predictor Variables: The depth (D) and its derivatives (Table 1) were used as predictor variables. Although RFs can handle a high number of predictor variables with similar information, the exclusion of highly correlated variables can improve the RF performance and decrease computation time (Che Hasan, 2014; Li et al., 2016). Thus, the correlation between derivatives was investigated using the Spearman's rank correlation coefficient. None of the variable pairs was




highly correlated ($\rho \geq 95$) and consequently, all of them were used for RF modelling (Appendix A).

Calibration of the model: During the calibration process, the RF parameters were adjusted as follows. The number of predictor variables to be randomly selected at each node (*mtry*), the minimum size of the terminal nodes (*nodesize*) and the

number of trees to grow (*ntree*) were set to the default values. For regressions *mtry* is 1/3 of the number of predictor variables (rounded down), *nodesize* is 5 and *ntree* is 500 (Liaw and Wiener, 2002). RF has demonstrated to be robust regarding these parameters and the default values have given trustworthy results (e.g. Liaw and Wiener, 2002; Diaz-Uriarte and de Andres, 2006; Cutler et al., 2007, Okun and Priisalu, 2007; Li et al., 2016 & 2017). With regards to the subsampling method (*replace*), the subsampling without replacement was selected. Although the initial implementations of the RF

algorithm use subsampling with replacement (Breiman, 2001a), later studies showed that this process might cause biased selection of predictor variables that vary in their scale and/or in their number of categories, resulting in a biased variable importance measurement (Strobl et al., 2007, 2009; Mitchell, 2011).  Based on recent findings, the raw variable importance was preferred (*unscaled*) as the final parameter (Diaz-Uriarte and de Andres, 2006; Strobl et al., 2008a, 2008b, 2009). Using these settings, the influence of the training sample size was examined (10 to 90% of the total sample in steps of 10%) and

compared based on the Mean of Squared Residuals (MSR) using the respective equation provided in the *randomForests* R package (Liaw and Wiener, 2002). The different training groups need to be considered as representative of the total sample, in order to capture the heterogeneity of the Mn-nodules' spatial distribution. The spatially random selection of subsamples by MGET ensured similar statistical characteristics in each group (Appendix A). After the optimal training sample size was defined, the influence of the number of growing trees (*ntree*) and the influence of the number of predictor variables to be

randomly selected at each node (*mtry*) was examined. In total, ten different *ntree* values (100 to 1000 in steps of 100) and seven different *mtry* values (1 to 7 in steps of 1) were tested. For each case of different training sample size, *ntree* and *mtry* parameter, the model was run ten times and the results are presented as the average value of these ten runs (Appendix A).

Selection and external validation of the optimal model: Based on the above-mentioned results and considering the sampling and computational cost, the optimal model was selected, run for 30 iterations and applied to the entire study area. Its

predicted values were validated with the observed values from the remaining dataset that was not used.

### 3.6 Resource Assessment

As the optimal RF model was applied to the entire Block G77, an estimate of the abundance (kg/m$^2$) was computed, based on the analogy between the corresponding abundance measured from the average number of Mn-nodules in the box corer data and the number of Mn-nodules/m$^2$ in each cell of the final result of the RF model. Considering that the collector can recover

buried Mn-nodules from a maximum depth of 10-15 cm (Sharma, 1993 and 2010), the ratio of 1.32 was applied to account for Mn-nodules not detected in the images, and areas with a slope of  >3° were excluded, assuming that a potential mining vehicle is limited to less steep slopes (UNOET, 1987).





## 4. Results

**4.1 Data Exploration**

The analysis of AUV photos with the CoMoNoD algorithm (Schoening et al., 2017a) revealed a rather heterogeneous pattern of Mn-nodules/m$^2$ in the study area showing adjacent areas with high and low Mn-nodules number (Figure 3a). The number of Mn-nodules/m$^2$ changes within less than 100m in the overall study area and in the two main sub-areas b and c (Figures 3a-c). This heterogeneity causes the large range of Mn-nodules/m$^2$ (Figure 3a). In half of the photos (48%), the number of Mn-

nodules/m$^2$ varies from 30 – 43 with the mean value being 36.6 Mn-nodules/m$^2$. The very small alternation of 5% trimmed mean values (Table 3) indicates the absence of extreme outliers, which is confirmed by box-plot analysis (Figure 4c). According to this plot, there are only 21 mild outliers (according to Hoaglin et al., 1986; Dawson, 2011), which correspond to the 0.18% of the total observation. This percentage is smaller than the 0.8% threshold that has been suggested for normal disturbed data (Dawson, 2011). The histogram of Mn-nodules/m$^2$ (Figure 4a) shows a good fit with the superimposed

theoretical normal curve, with most of the frequency counts in the middle and the counts die out in the tails. The shape of the distribution is rather symmetrical, this fact is approved by the equal mean and median and the slightly different mode (Table 3). Similarly, the visual inspection of the probability plot (Figure 4b) shows a good match as a linear pattern is observed for the greatest part, with slight deviation existing only in the outer parts of the curve. The small values for skewness and kurtosis combined with the large sample size further support the normal distributed pattern of the data (Table 3). Especially

for large data samples, measurement of skewness and kurtosis combined with the visual inspection of histogram and probability plot are recommended ways of examining normality of data (D' Agostino et al., 1990; Yaziki and Yolacan, 2007; Field, 2009; Ghasemi and Zahediasl, 2012; Kim, 2013). Presence of normality in data is not a prerequisite assumption in order to perform the RF (Breiman, 2001a); as it is with geostatistical interpolation techniques like as kriging (e.g. Kuhn et al., 2016). Nevertheless, this examination give us a better understanding of the Mn-nodules' distribution inside the study

area, while it is an important step in order to examine potential extreme observations which may be derived from wrong measurements and could artificially change the training range during RF predictive modelling. Moreover, absence of linear correlation was observed between Mn-nodules/m$^2$ and the produced derivatives, indicating the complexity of the phenomenon (Appendix B).

**Table 3.** The descriptive statistics of the number of Mn-nodules/m².

|  | Mean | 5% Trim. Mean | Median | Mode | SD | Min. | Max. | Skew. | Kurtosis |
|---|---|---|---|---|---|---|---|---|---|
| Mn-nodules/m² | 36.6 | 36.4 | 36.4 | 39 | 9.2 | 6.8 | 78.2 | 0.1 | -0.4 |






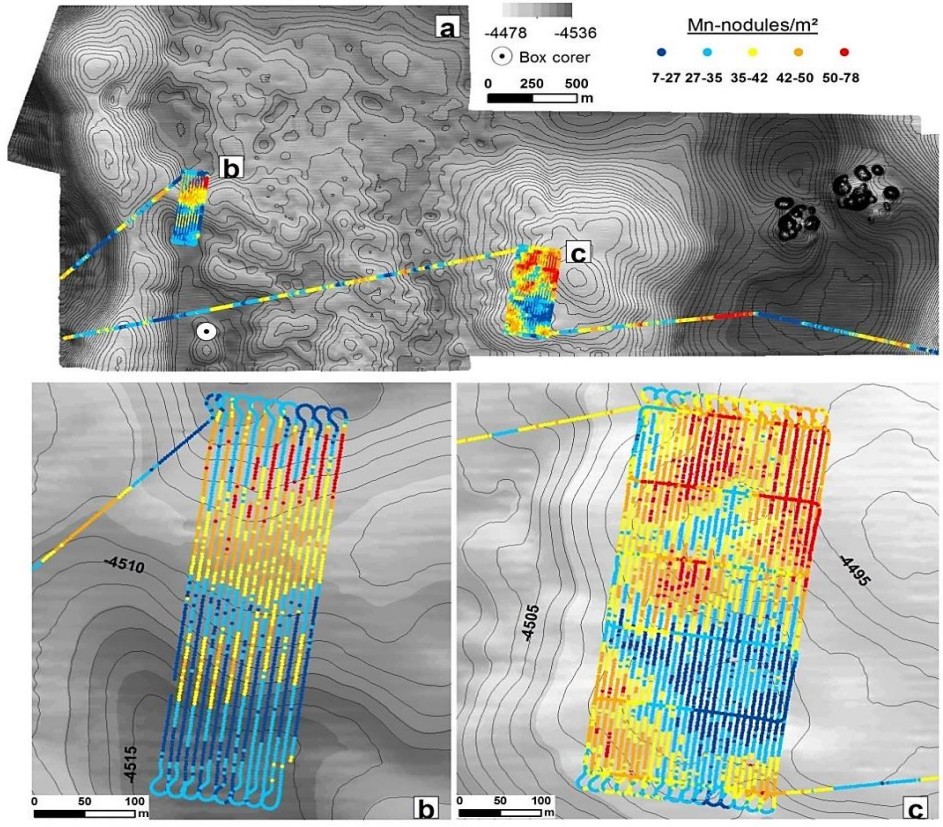

**Figure 3.** a) The spatial distribution of Mn-nodules/m$^2$ inside block G77 and the box corer position. b) The spatial distribution of Mn-nodules/m$^2$ inside the sub-area b. c) The spatial distribution of Mn-nodules/m$^2$ inside the sub-area c.

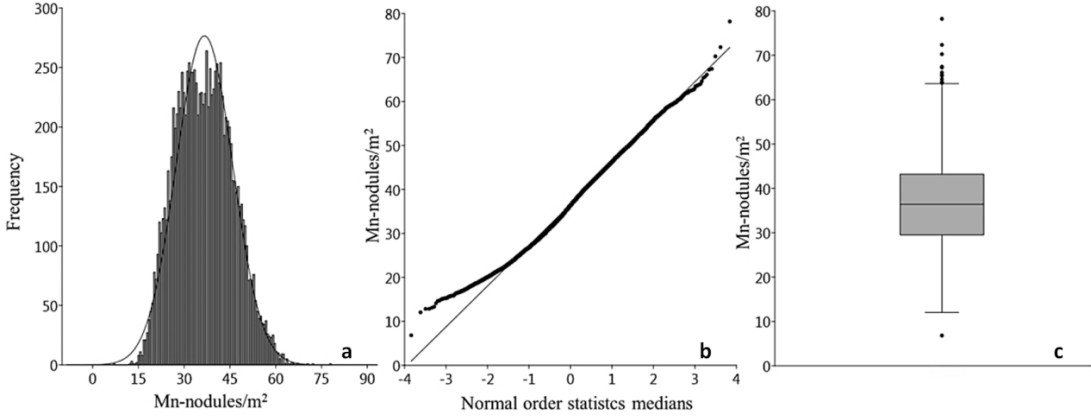

**Figure 4.** a) Histogram of Mn-nodules/m$^2$ with the superimposed normal curve. b) The normal probability plot of Mn-nodules/m$^2$. c) The box plot of Mn-nodules/m$^2$.



## 4.2 Spatial Analyses

Spatial analyses revealed the presence of a spatial autocorrelation in the distribution of Mn-nodules/m$^2$. The GMI, with I=0.6989, p<0.01 and Z-score>2.58 indicates a positive spatial autocorrelation. According to the incremental analysis, the
index takes its highest value in the first 50m with a gradual decrease, approaching 0 values after 400m distance (Figure 5a). Similarly, the results from the LMI show that the main size of the spatial clusters does not exceed 400m in either direction (Figure 6a). The main types of these clusters are H-H and L-L groups (Table 4 & Figure 6a). A distinct 'buffer/transitional zone' with Mn-nodules was found between these two clusters, which does not show a significant autocorrelation (Figure 6b & 6c). Approximately one-third of the data does not have a significant clustering (NS). In the sub-area c, in the outer parts of
these zones without significant spatial clustering, the few local H-L and L-H groups are located. Both H-L and L-H (from the entire study area) only account for 2.1% of the data (Table 4). The comparison of the number of Mn-nodules/m$^2$ between the groups shows a clear discrimination between H-H and L-L clusters (Figure 5b). The H-H clusters are in areas with 37.9-78.2 Mn-nodules/m$^2$ whilst the L-L clusters are in areas with 6.8-35.2 Mn-nodules/m$^2$.

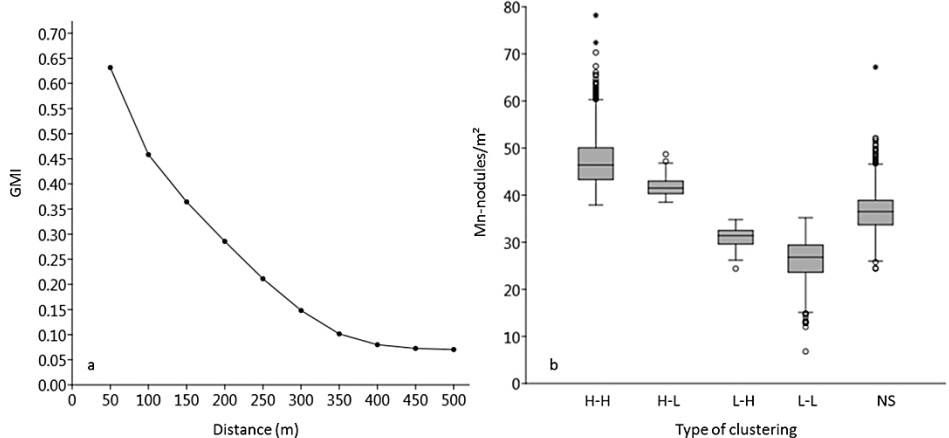

**Figure 5.** a) The GMI decrement due to increasing distance, after the first 50m. b) The range of Mn-nodules/m$^2$ in each clustered group.

**Table 4.** Number and % percentage of samples in each type of spatial clustering.

| Cluster Type | H-H | H-L | L-H | L-L | NS |
|---|---|---|---|---|---|
| Counts (n) | 3472 | 121 | 113 | 3523 | 4047 |
| Counts (%) | 30.8 | 1.1 | 1.0 | 31.2 | 35.9 |





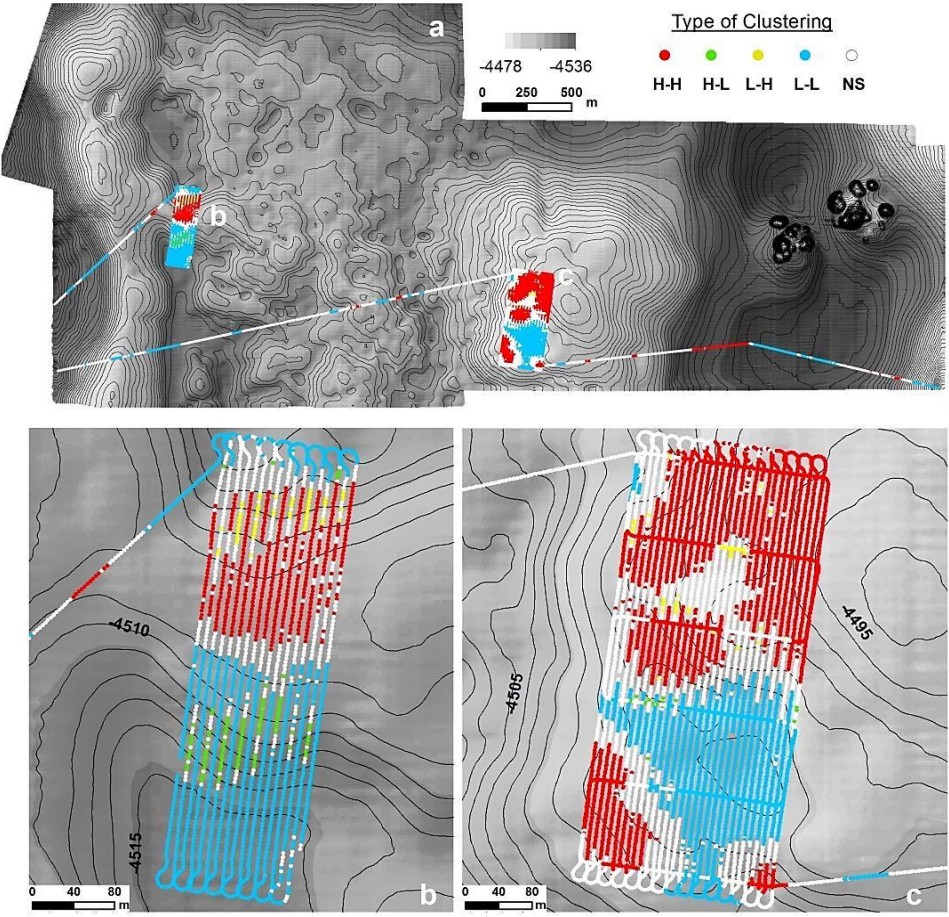

**Figure 6.** a) The spatial distribution of the significant cluster types inside the block G77. b) The spatial clusters inside the sub-area b. c) The spatial clusters inside the sub-area c

The application of the LMI reveals a bias that exists in the data due to the sampling procedure, especially in the sub-area b (Figure 6b). Here, the presence of the slope around 2.8° forced the AUV to vary its altitude between the ascending and descending phase (Figure 7b). This variation seems to affect the image quality resulting in counting fewer nodules for higher altitudes of the AUV (Figure 8 & 9). This is also confirmed by the distribution map of the Mn-nodules/m² (Figure 3b). It is important to emphasize that this difference clearly shows up in the LMI results (Figure 6b) and not in the distribution map (Figure 3b); here the arbitrary choice of color scale can hide this bias during plotting. The comparison of the detected Mn-nodules/m² in these adjacent lines, inside the small sub-area b, gives a ratio ≈1.4 between photos that have been acquired in 7-9 m altitude and those in 9-11 m altitude. The ratio is higher (≈1.8) between photos from 5-7 m and 9-11 m altitude. In contrast, the ratio between photos from 5-7 m altitude and those in 7-9 m altitude is ≈1.25 indicating that the problem mainly exists in upper and lower flying altitudes. Despite their different ratio, none of these groups contain extreme high or low





values of Mn-nodules/m². Moreover, in several parts of the block, the photos from higher altitude are the only source of information without the ability for further comparison and consequently, they cannot be excluded from the modelling procedure.

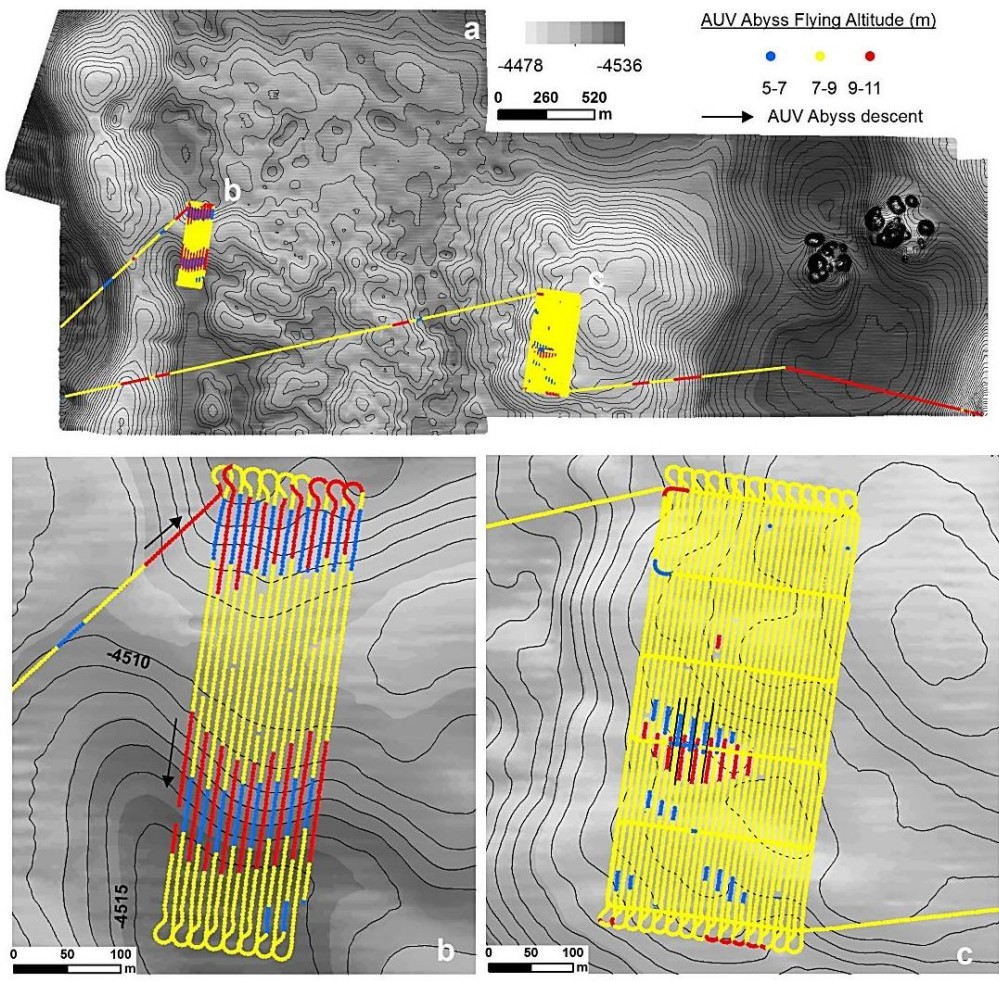


**Figure 7.** a) The altitude of AUV Abyss inside block G77. b) The altitude inside the small sub-area b, where the presence of the slope forces the AUV to modify its altitude, flying closer to the seafloor in the ascending phase (blue lines) and farther from the seafloor in the descending phase (red lines). c) In the big sub-area c, the AUV flying altitude is mainly constant between 7-9 m for the entire part.




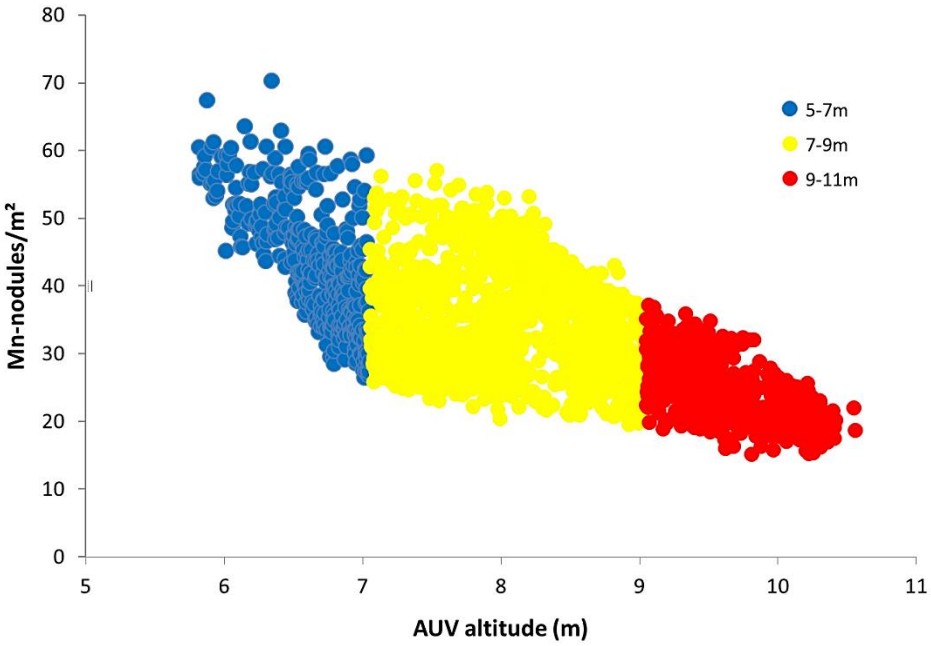

**Figure 8.** Scatterplot of the AUV altitude (m) and the estimated number of Mn-nodules/m² inside sub-area b. The colours correspond to the colour scale in figure 7.

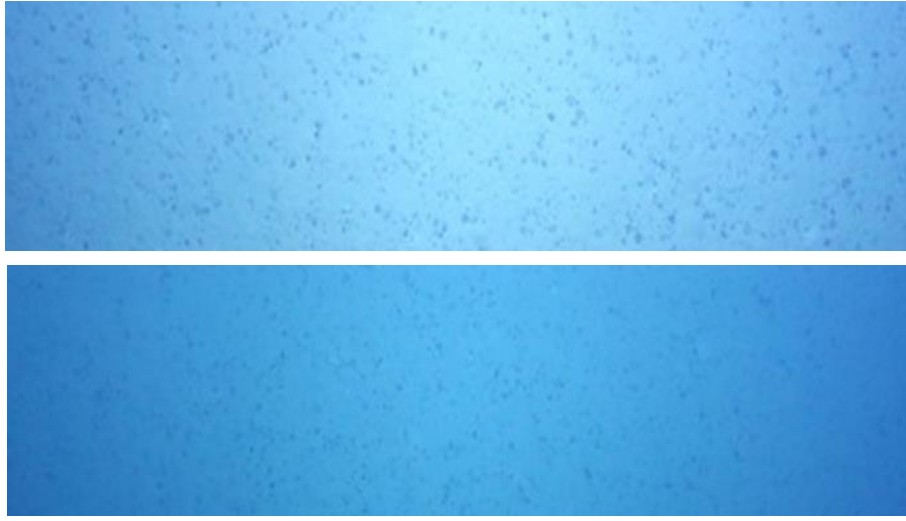


**Figure 9.** Adjacent AUV photos from consecutive dive tracks that have been obtained inside sub-area b, from: a) lower (5-7m) and b) higher (9-11m) altitude. Notice the decrement in the image brightness. (The area of the photos represent the central part of the photo, ca.1/4 of the original photo size).





Spatial distribution of median size: Plotting of the median size in cm$^2$ (Figure 10) showed that the number of Mn-nodules/m$^2$
is anti-correlated to the median Mn-nodule size. The Spearman's rank correlation coefficient and R$^2$ between these two
variables are -0.50 and 0.25 respectively, supporting this observation (Figure 11a); other studies found similar results
(Okazaki and Tsune, 2013; Kuhn and Rathke, 2017; Peukert et al., 2018a). The box plot analysis of the median size values
between the H-H and L-L clustered groups showed that although the L-L group contains the entire range of median size
values (2.8 to 15.9 cm$^2$), the H-H group does not contain values above 10cm$^2$ (2.7-10cm$^2$). This means in consequence that in
areas with significant clustering of higher numbers of Mn-nodules/m$^2$ the size of Mn-nodules tends to be smaller (Figure
11b).

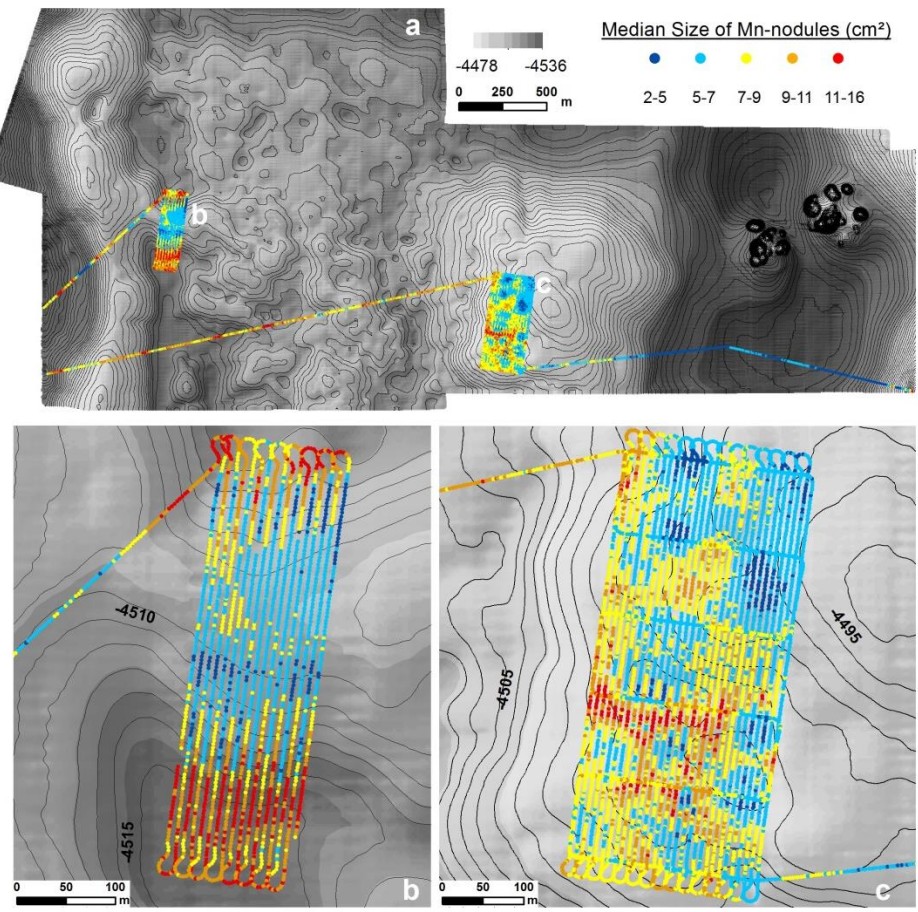

**Figure 10.** a) The spatial distribution of median Mn-nodule size (in cm$^2$). b) The estimation of median Mn-nodule size in
sub-area b and mainly in its southern part has been probably affected by the non-constant altitude of the AUV. c) The
distribution of the median size inside sub-area c shows a clumped pattern, too.





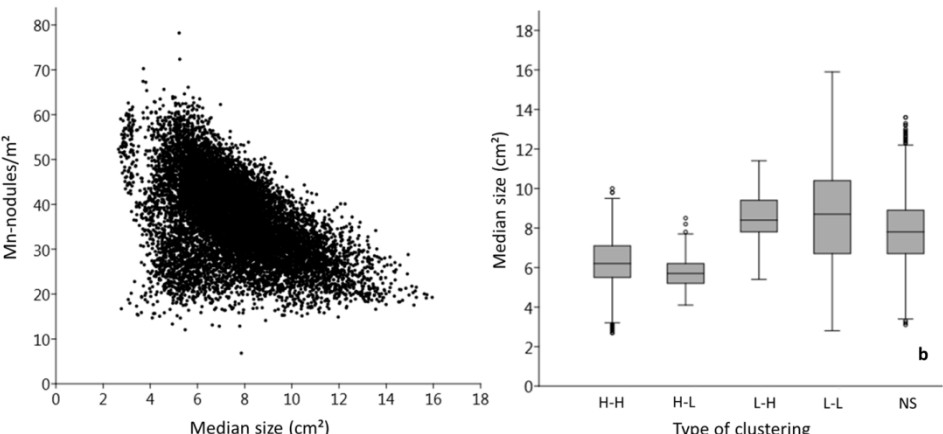

**Figure 11.** a) The plot of median size (cm$^2$) and number of Mn-nodules/m$^2$. e) The range of median size (cm$^2$) in type of cluster. Notice the distinct difference in the range between the H-H and L-L cluster type.

## 4.3 RF Predictive modelling

### 4.3.1 Effect of training sample size, *ntree* and *mtry* parameter

The results of the modelling procedure demonstrate that the RF algorithm is influenced by the size of the training sample (Figure 12a). This finding is in accordance with other studies, in which larger training samples tended to increase the performance of RF (Li et al., 2010 and 2011b; Millard and Richardson, 2015). The inclusion of a more representative range of the observed values and consequently larger spectrum of the causal underlying relationships, assist the RF to build a better model for the prediction of the value distribution inside the study area. For our data the decrement becomes smaller when the size of the training sample increases further; it reaches a minimum value of 0.2 between 80% and 90%, showing that these additional 10% do not notably benefit the RF model. However, the absence of stabilization of the error to a minimum value indicates that more optical data are needed from this block. The small decrement in error between 80% and 90% was the decisive factor to select 80% of the data as training samples (also considering the larger number of remaining validation data and the reduced computational effort). Based on this dataset, the examination of different numbers of trees showed that the RF error remains constant after 600 trees (Figure 12b). Less trees result in a larger error; this particularly becomes evident with less than 300 trees. With more than 300 trees the range of the error is reduced (Appendix B). A higher number of trees enables higher *mtry* values as there are more opportunities for each variable to occur in several trees (Strobl et al., 2009). Similarly to the *ntree* parameter, a larger number of *mtry* values results in a reduced error (Figure 12c). The error reaches a minimum and cannot be reduced further for *mtry* = 6; with values below 3 the error increases significantly. The different numbers of *ntree* reduced the error by only 0.6 in the MSR (from 18.8 to 18.2), in contrast different *mtry* values reduced the error by 5.8 in the MSR (23.4 to 17.6), highlighting its importance for the prediction accuracy. In general a higher number of




*mtry* values is suggested for RF studies with correlated variables to result in a less biased result regarding the importance of
each variable; this is because the higher number increases the competition between highly correlated variables, giving more
chances for different selections (Strobl et al., 2008a). The finally selected *mtry* value of 6 coincides with the recommended
approach for *mtry* (default, half of the default, and twice the default) suggested by Breiman (2001a). Albeit the importance of
this analysis, within the model with 80% of the data as training sample, the decrease in error by the use of RF tuned values
instead of RF default values was only 0.7 in the MSR values, whilst the greatest reduction in error (16.5 in the MSR values)
came from the increase in training data set size. This highlights the increased sensitivity of the method with respect to
training data and that the recommended settings in the R *randomforest* package (Lia and Wiener, 2002) give trustworthy
results, increasing its simplicity and operational character.

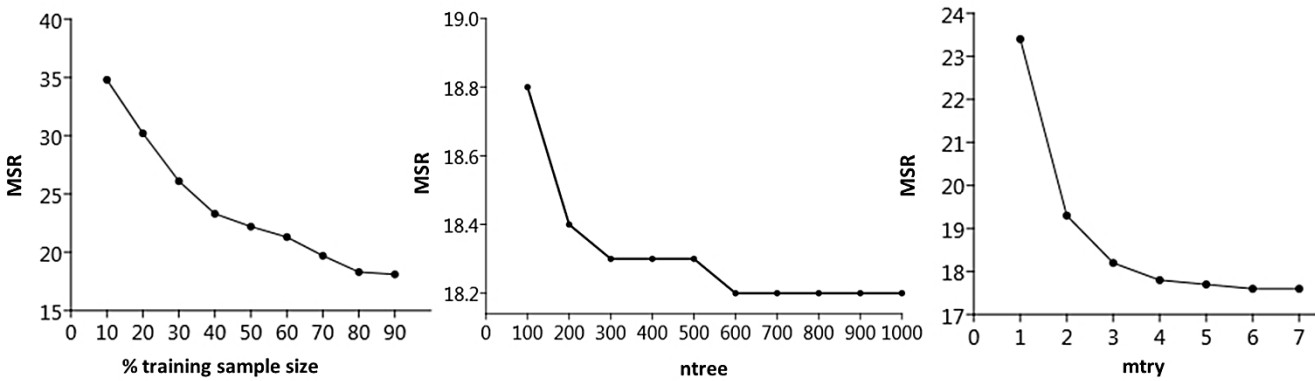

**Figure 12.** a) The effect of training sample size in RF error (in MSR). b) The effect of *ntree* parameter in RF error (in MSR).
c) The effect of *mtry* parameter in RF error (in MSR).

### 4.3.2 Selection, application and external validation of the optimal model

Based on the above-mentioned findings, the optimal RF regression model which uses 80% of training data, 600 trees and 6
predictor variables to be randomly selected at each node, was selected and applied to the entire block G77. The comparison
of the predicted values with the observed values from the remaining 20% (2,255 observations) of validation data showed a
good predictive performance (Table 5). Analytically, MAE and RMSE have very low values, $R^2$ has a high value and both
Pearson's and Spearman's correlation coefficients show a strong positive correlation between the predicted and observed
values. The small deviation between MAE and RMSE and the same good correlation of the Pearson and Spearman factor
point towards the absence of extremely high or low predicted values (outliers). Moreover, the performance is rather stable
among all the iterations (Appendix B).

**Table 5:** The values of validation measures between predicted and observed data.



| MAE | MSE | RMSE | R² | Pearson | Spearman |
|------|------|------|-----|---------|----------|
| 3.1 | 19.0 | 4.4 | 0.8 | 0.9 | 0.9 |

The scatterplot and box plot (Figure 13a and 13b) illustrate this good match between predicted and observed values, as confirmed also by the descriptive statistics, which have almost equal mean, median, skewness and kurtosis values (Table 6). The residual analysis confirmed further the robustness of the model (Appendix B).

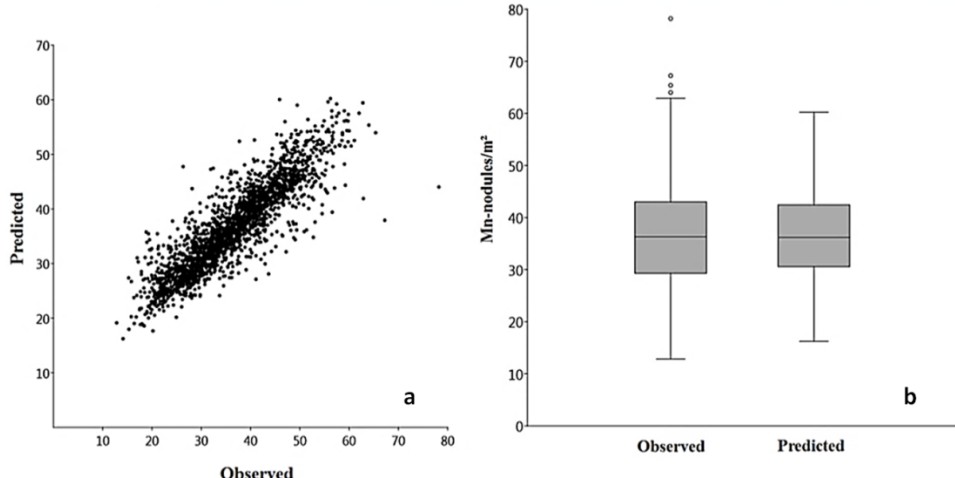


**Figure 13.** Comparison between observed and predicted values: scatterplot (left) and box-plots (right)

**Table 6.** Descriptive statistics of observed and predicted values

|  | Mean | Std. Error | 5% Trim. Mean | Median | Mode | SD | Min. | Max. | C.L (95%) |
|------|------|-----------|---------------|--------|------|-----|------|------|-----------|
| Observed | 36.5 | 0.2 | 36.3 | 36.3 | 40.8 | 9.4 | 12.8 | 78.2 | 0.4 |
| Predicted | 36.7 | 0.2 | 36.5 | 36.2 | 33.9 | 7.8 | 16.2 | 60.2 | 0.3 |

The statistical analysis also reveals the limitations of the RF model which cannot predict beyond the range of training values. It underestimates the maximum predicted values and overestimates the minimum values (Figure 13b & Table 6), a limitation

also mentioned by other authors (e.g. Horning, 2010). This happens, because in regression RF the result is the average value of all the predictions (Breiman, 2001a).

### 4.3.3 RF predicted distribution of Mn-nodules/m²

The final application of the RF model for the entire block G77 predicts that the majority of the area is covered by 30-45 Mn-nodules/m² (Figure 14). In the central-western part the distribution is quite uniform (at this scale) with few small areas of





lower numbers. In the western part, there are two extended areas along the base of the hill with the lowest number of Mn-nodules/m². Both of these areas have a linear shape in N-S direction and follow the seafloor topography with increased slope (>3°). The third main patch with minimum Mn-nodules/m² occurs in the eastern depression part. In contrast, areas of higher number of Mn-nodules/m² are located mainly in the central upper part of the hill and eastward facing slope of eastern depression and south of the sub-recent hydrothermally active area.

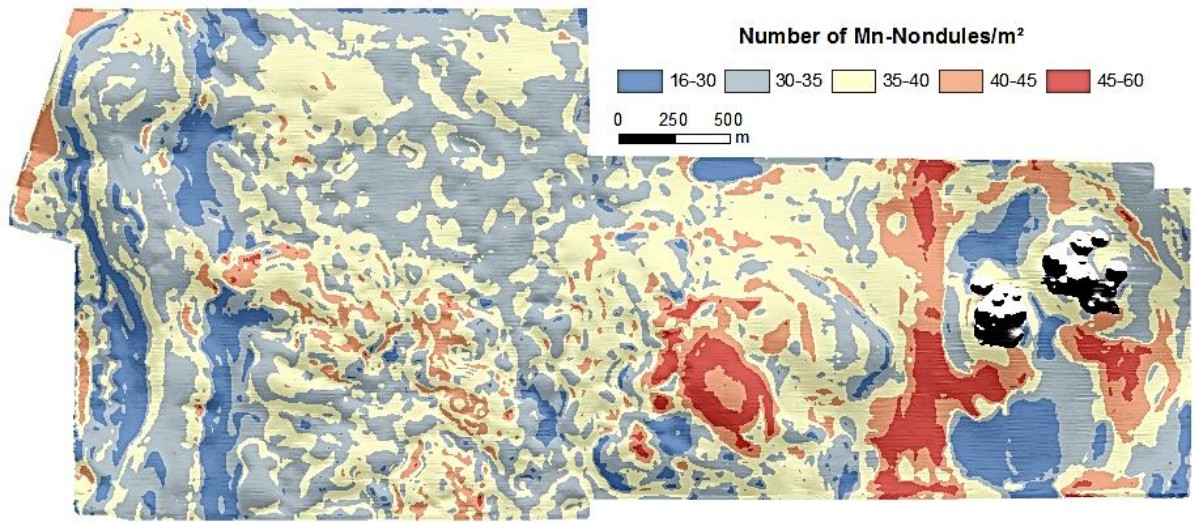


**Figure 14.** The RF predicted distribution of Mn-nodules/m² inside block G77.

### 4.3.4 RF importance

The analysis of the RF importance showed that the best explanatory variable for the distribution of Mn-nodules/m² is depth (Figure 15a). The partial dependence plot of depth shows that there are specific ranges of the depth, which promote higher
numbers of Mn-nodules/m² aggregated in a nonlinear way (Figure 15b). Such nonlinear relationships between predictor and response variables have already been described in the past, both in marine (e.g. Zhi et al., 2014; Li et al., 2017) and terrestrial environments (e.g. Cutler et al., 2007). The following two most important variables are the TPI_B and TPI_M. TRI, TPI_F, C, and S follow in importance (Figures 15a). All of them contribute in a nonlinear way, too (Appendix B). Pl.C and Pr.C do not contribute significantly as explanatory variables in the performance of the RF model (Figure 15a and
Apendix B). Although the RF demonstrates good overall performance, the small study area and the arbitrary choice of the spatial scales for the TPI and other derivatives, limit the potential of these variables as indicative explanatory variables on a broader scale. It is well established that surface derivatives are scaled-depended with different analysis scales to create alterations in results. Thus the combined use of different scales (here TPI) in the analysis and modelling procedure can produce models that do capture the natural variability and scale dependence (Wilson et al., 2007; Miller et al., 2014;  Ismail
et al., 2015; Leempoe et al., 2015).





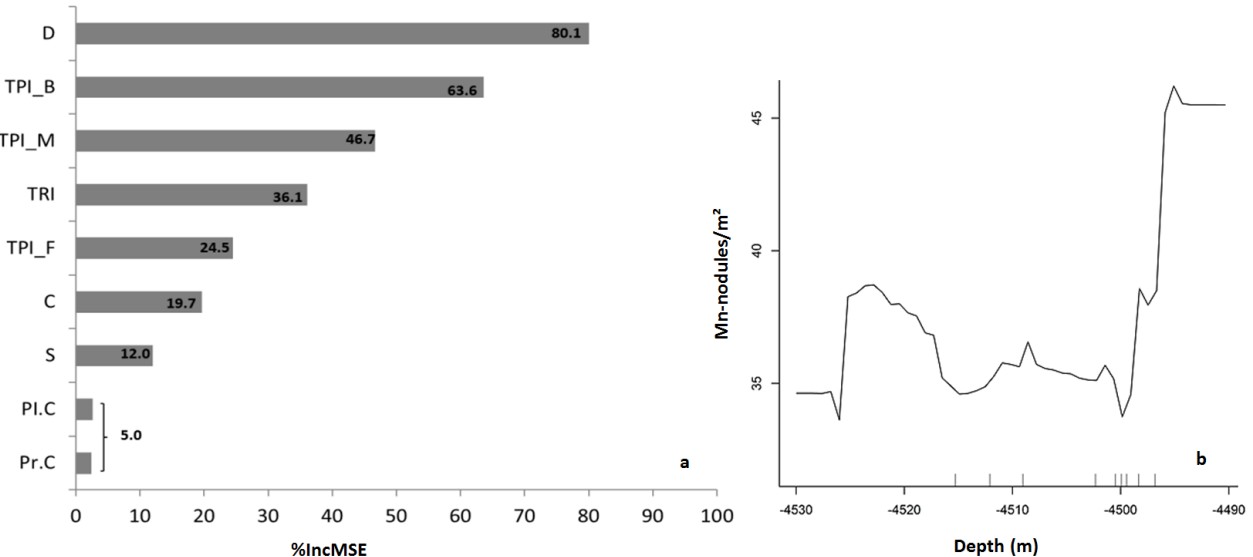

**Figure 15:** a) The variable importance of each predictor in the RF model. b) The partial dependence plot of the Depth (right). The ticks inside the graphs indicate the deciles of the data.

### 4.3.5 Estimation of abundance (kg/m²) of Mn-nodules

The predicted Mn-nodule distribution was combined with the abundance from box corer data (and corrected with the ratio between buried/unburied Mn-nodules, in order to include the top ~15 cm of the sediment), resulting in the Mn-nodules' abundance map shown in Figure 16. According to this map, block G77 is a promising area for mining operations. The entire block is above the cut-off abundance of 5 kg/m2 (UNOET, 1987), with a mean value of 33.8 kg/m2. We calculated that 84% of block G77 has slopes below 3°, steeper slopes are located mainly at the outer parts of the block, a fact that would ease

establishing an ideal mining path. In this respect, the AUV-scale mapping provides vital information for a potential mining path by decreasing the possibility of machine failure due to poorly mapped steep slopes not detected e.g. by ship-based bathymetry (Peukert et al., 2018b). Mn-nodule distribution maps with this resolution increase the mining efficiency because local deposit variations can significantly affect the performance of the pick-up rate, which is likely determined by technical parameters of the mining vehicle as well as the size, burial depth and abundance of Mn-nodules in the seafloor (Chung,

1996). The exclusion of areas with slopes > 3° resulted in 8 km² mineable seafloor surface. Assuming a constant 80% collection efficiency (Volkmann & Lehnen, 2018) and a 30% reduction of the Mn-nodule weight by removal of water (Das & Anand, 2017), the dry mass of Mn-nodules that can be extracted from the surface and the first 15 cm of the sediment column amounts to ca. 190,000 t. In a back-of-the-envelope calculation this quantity, assuming constant metal content inside the study area , equal to the average metal concentrations inside the CCZ (Table 7) (Volkmann, 2015), and 90% metal

recovery efficiency; could result in an estimated resource haul of 45,450 t Mn, 2,232 t Ni, 1891 t Cu, 374 t Co, and 102 t



Mo (Table 7).

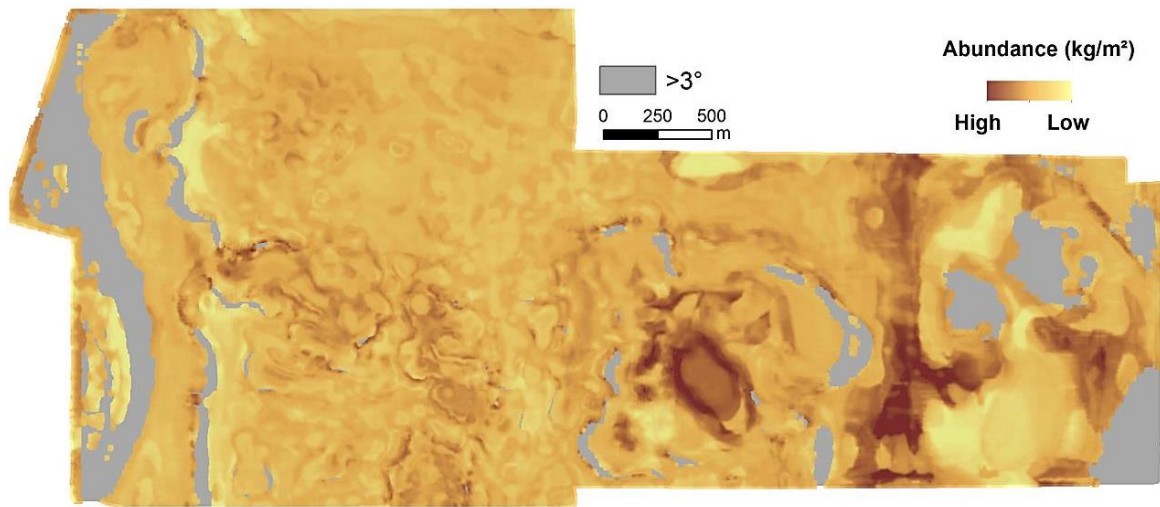

**Figure 16:** The total abundance of Mn-nodules from the surface and embedded in the sediment (max. 15 cm), in areas with slope ≤ 3° inside block G77 (continuous values of abundance are not given due to confidentiality).

**Table 7.** The estimated amount of metal mass for 5 metals, based on the average values of metal content inside CCZ and a 5metal HCl-leach recovery method (Volkmann, 2015).

| Total Wet Mass (t): | 270,400 | | | | |
|---|---|---|---|---|---|
| Total Dry Mass (t): | 189,280 | | | | |
| Metal Content | Mn | Ni | Cu | Co | Mo |
| wt%: | 26.68 | 1.31 | 1.11 | 0.22 | 0.06 |
| Equal to (t): | 50,500 | 2,480 | 2,101 | 416 | 113 |
| 90% metal recovery (t) | 45,450 | 2,232 | 1,891 | 374 | 102 |

## 5. Discussion

We present a case study that highlights the applicability of the combination of AUV bathymetric and optical data for Mn-nodules resource modelling using RF machine learning. The use of AUVs for collecting hydroacoustic and optical data in areas of scientific and commercial interest can provide more precise bathymetric and Mn-nodules distribution maps. Regarding the bathymetric maps, the accurate and detailed reconstruction of the seafloor bathymetry in meter-scale resolution enables to use bathymetry and its derivatives as source data layers within a high-resolution RF model. High-quality bathymetric data, as primary model explanatory variables, is a-priori step, as the occurrence of acquisition artefacts





may affect the robustness of the modelling procedure (Preston, 2009; Herkül et al., 2017). The combined use of cameras as
the DeepSurveyCamera (Kwansnitschka et al., 2016) for acquiring high-resolution photographs, and an automated analysis
with a state-of-the-art algorithm (Schoening et al., 2017a) provides essential quantitative information about the distribution
of Mn-nodules. Image analysis results are more robust for constant AUV altitudes (7-9m) above flat areas (<3°), while the
alternation of the flying altitude and camera orientation during the ascending & descending phases limit the quality of the
obtained images and can affect the derived number of Mn-nodules/m². Inside block G77, the number of Mn-nodules/m$^2$
seems to follow a normal distribution without extreme outliers and without being linearly correlated with the used predictor
variables. Spatially, a clumped autocorrelated pattern is demonstrated, mainly with clustered areas of H-H and L-L values.
Whether this heterogeneity is caused by external processes (e.g. topographic characteristics, geochemical conditions,
availability of nucleus material etc.) or reflects the interaction of neighboring Mn-nodules is still unclear. The negative
correlation between the number of Mn-nodules/m$^2$ and the median Mn-nodule size implies that may higher numbers of Mn-
nodules could provide more fragments as potential nucleus material at less available space and oxides for individual Mn-
nodule growth. Opposite to this, a recent study from Kuhn and Rathke (2017) showed that the blanketing of the Mn-nodules
by sediments is higher for larger Mn-nodules and, as a result, fewer large nodules will be counted; resulting in biased results
regarding the areas with bigger size of Mn-nodules. Probably, all of these effects can happen at the same time (with different
degrees of influence) promoting a given, scale-dependent spatial structure. This study did not consider geochemical
properties of the sediments as input data in the modelling process, which might give additional clues to why Mn-nodules are
distributed as they are. However, RF importance and partial dependence plots show that bathymetric factors tend to affect
this distribution in a non-linear way and with the bulk of data plotting in specific ranges of the bathymetric derivatives. It
should be acknowledged that the aim of any ML predictive model is to derive accurate predictions based on an existing
(large) number of measurements, to capture a complex underlying relationship (e.g. non-linear and multi-variate) between
different types of data, for which our theoretical knowledge or conceptual understanding is still under development
(Schmueli, 2010; Lary et al 2016). Especially due to the constantly increasing size of scientific multivariate data in marine
sciences, ML and RF are considered important analytic tools that can objectively reveal patterns of a (unknown)
phenomenon (Genuer et al, 2017; Kavenski et al, 2009; Lary et al 2016). Such predictions may be used to derive causalities
or may drive the creation of new hypotheses. In other words, for a predictive model the 'unguided' data analyses come first
and the interpretation follows (Breiman, 2001b; Schmueli, 2010; Obermeyer and Emanuel, 2016). The RF modelling takes
advantage of the multi-layer information and is able to handle non-linear and complex relationships between image-derived
Mn-nodule data and explanatory variables from hydro-acoustics while being resistant to overfitting (Breiman, 2001a).
Moreover, the randomization of the input training points in each tree in each run, resulting in a complete different training
dataset each time with mixed points from the entire study area. This random selection and mixing of points, is appropriate
for clustered data, as ignores their spatial locations and consequently limits the influence of spatial autocorrelation.  To this
direction several authors, have included the values of latitude/longitude and even the LMI values as predictor variables in
order to increase the model performance (e.g. Li, 2013; Li et al., 2011b; Li et al., 2013). RF has a high operational character





due to its relatively simple calibration, which does not request extensive data preparation/transformation or need for geostatistical assumptions (e.g. stationarity). RF model runs can easily be implemented inside various software packages and

its increased stability can even allow a small number of iterations to compute sufficient results (Cutler et al., 2007; Millard and Richardson, 2015). The examination of the main two tuning parameters (*ntree* and *mtry*) showed that the model performance can be increased compared to default values. However, the largest improvement results from using more training data. In this respect, more photos would potentially improve the RF performance as no clear threshold was observed. Although the number of 11,276 photos seems to represent a large data set, the heterogeneity of the distribution and the

occurrence of spatial clusters (patches) in different sizes and the inherent need of RF and ML in general for big training datasets (van der Ploeg et al, 2014; Obermeyer and Emanuel, 2016), stresses the need for collecting more and well distributed data. Although the exact reasons for the patchy distribution are not fully understood, the distribution pattern is essential for planning box corer sampling. A random spatial sampling reduces the possibility of dependence among observations in a homogeneously distributed population (Cochran 1977), but it is not appropriate for clustered populations as

it cannot eliminate autocorrelation between neighboring sample locations that are inside the spatial influence of the underlying phenomenon (Legendre and Legendre, 1998). In other words, if two or more box-corers are obtained from the same patch, the results will still not be representative for the entire study area. Thus, the deployment of box corers should be executed only after the acquisition, processing and spatial analysis of bathymetric and optical data. The number of box corers should be the maximum feasible, with at least one within each main patch. Moreover, sampling should occur in locations

with available photos (to enable direct comparison) and locations without photos for a true validation of the RF prediction. In other words, the optical data acquisition should be guided by the bathymetric and backscatter seafloor characteristics, and be followed by box core sampling that targets all defined 'seafloor-classes' by considering direct correlations with the previously gathered optical and hydro-acoustic data. This 'priori' knowledge of the distribution of the Mn-nodules number and size in such scale can contribute to the biological data survey planning, too. Recent studies showed that the abundance

and species richness of nodule fauna inside the CCZ is affected by the abundance of Mn-nodules (Amon et al., 2016; Vanreusel et al., 2016) as long as from the size of them (Veillette et al., 2007). Thus, based on the needs of studies the results of optic data and RF modelling can lead the data sampling in high priority areas (e.g. these with highest commercial interest).Finally, the resource assessment showed that block G77 is a potential mining area with high average Mn-nodules density and gentle slopes. While here the threshold of 3° (UNOET, 1987) was used, newer plans for mining machines seem

to enable operations on steeper slopes (Atmanand and Ramadass, 2017) increasing the total amount of collected Mn-nodules within the herein considered area.

## 6. Conclusions

The results of this study show that the acquisition and analysis of optical seafloor data can provide quantitative information on the distribution of Mn-nodules. This information can be combined with AUV-based MBES data using RF machine



learning to compute predictions of Mn-nodule occurrence on small operational scales. Linking such spatial predictions with sampling based physical Mn-nodule data provides an efficient and effective tool for mapping Mn-nodule abundance.

**Competing interests:** The authors declare that they have no conflict of interest.

**Special issue statement:** This article is part of the special issue "Assessing environmental impacts of deep-sea mining –
revisiting decade-old benthic disturbances in Pacific nodule areas". It is not associated with a conference.

**Acknowledgments:** We thank the captain and crew of RV SONNE for their contribution to a successful cruise. We express our gratefulness to the GEOMAR AUV team for their support during the cruise. We thank Anja Steinführer for pre-processing of the AUV MBES data and Mareike Kampmeier for advice during the post-processing analysis of the MBES
data. We thank Inken Preuss for proof-reading the manuscript. Finally, we thank the GEOMAR Library team for its support in gathering the necessary bibliography. All data were acquired within the framework of the JPIO Project "Ecological Aspects of Deep-Sea Mining", funded through BMBF-Grant 03F0707A. This is publication 35 of the DeepSea Monitoring Group at GEOMAR Helmholtz Centre for Ocean Research Kiel.


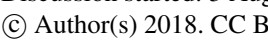



## Appendix A: Methodology

3.1 Hydro-acoustic Data Acquisition & Post Processing: The calculation of the bathymetric derivatives was performed with the SAGA GIS v6.3.0 Morphometry library (http://www.saga-gis.org/saga_tool_doc/6.3.0/ta_morphometry.html).

3.4 Spatial Statistics: Global Moran's I and Local Moran's I were performed with the ArcMap™ 10.1 software, using the Spatial Statistic Toolbox, according to its provided equations. As a null hypothesis, it is assumed that the examined attribute
is randomly distributed among the features in the study area. For the optimal conceptualization of spatial relationships, the Inverse Euclidian Distance Method was selected, as it is appropriate for modelling processes with continuous data in which the closer two samples are in space, the more likely they are to interact/influence each other or have been influenced from the same reasons. The distance threshold was set at 50m and the increment analysis was performed with a step of 50m. Moreover, the spatial weights were standardized in order to minimize any bias that exists due to sampling design (uneven
number of neighbors). Apart from the index value, the p-value and z-score are also provided. The Local Moran's I indicates statistically significant clusters and outliers for a 95% confidence level. The high number of observations (>>30) that was used ensures the robustness of the indexes.

3.5 RF Predictive modelling: Correlation among the derivatives was checked by Spearman's correlation coefficient ($\rho$). This coefficient was preferred due to the skewed distribution of the values in the derivatives. The majority of the possible pairs is
uncorrelated or weakly correlated. Only C vs. TPI_F and TRI vs. S have a strong correlation. However, they should not be excluded as they express different topographic characteristics and they are not correlated with the remainder of derivatives.

**Table A1.** Spearman's correlation coefficient for each pair of predictor variables

|  | D | S | Pl.C | Pr.C | TPI_B | TPI_M | TPI_F | C | TRI |
|---|---|---|---|---|---|---|---|---|---|
| **D** |  |  |  |  |  |  |  |  |  |
| **S** | -0.07 |  |  |  |  |  |  |  |  |
| **Pl.C** | 0.06 | -0.02 |  |  |  |  |  |  |  |
| **Pr.C** | 0.08 | -0.01 | 0.37 |  |  |  |  |  |  |
| **TPI_B** | 0.76 | -0.09 | 0.13 | 0.16 |  |  |  |  |  |
| **TPI_M** | 0.36 | -0.06 | 0.20 | 0.27 | 0.72 |  |  |  |  |
| **TPI_F** | 0.23 | -0.05 | 0.33 | 0.41 | 0.47 | 0.77 |  |  |  |
| **C** | -0.30 | 0.05 | -0.25 | -0.34 | -0.54 | -0.79 | -0.90 |  |  |
| **TRI** | -0.10 | 0.91 | -0.02 | -0.03 | -0.12 | -0.06 | 0.04 | 0.05 |  |

The 9 training samples with different size were created by the MGET tool: Randomly Split Table into Training and Test *Records*. The spatial randomness of the procedure, combined with the many available data resulted in training samples with



similar descriptive statistics.

**Table A2.** Descriptive Statistics of different training samples

| % Training Sample: | 10% | 20% | 30% | 40% | 50% | 60% | 70% | 80% | 90% |
|---|---|---|---|---|---|---|---|---|---|
| Training set size | 1127 | 2255 | 3383 | 4511 | 5638 | 6766 | 7894 | 9021 | 10148 |
| Mean | 36.5 | 36.3 | 36.6 | 36.6 | 36.6 | 36.7 | 36.6 | 36.7 | 36.6 |
| Std. Error | 0.3 | 0.2 | 0.2 | 0.1 | 0.1 | 0.1 | 0.1 | 0.1 | 0.1 |
| Std. Deviation | 9.3 | 9.2 | 9.4 | 9.2 | 9.2 | 9.3 | 9.3 | 9.2 | 9.3 |
| Minimum | 7 | 13 | 12 | 13 | 12 | 14 | 7 | 7 | 7 |
| Maximum | 63 | 70 | 72 | 66 | 78 | 78 | 78 | 72 | 78 |

**Appendix B: Results**

4.2 Data Exploration: A potential linear correlation between depth, bathymetric derivatives, and number of Mn-nodules/m$^2$ was investigated using the Spearman's rank correlation coefficient ($\rho$) because of the skewed distribution and presence of extreme values in the depth and bathymetric derivative values (Mukaka, 2012).

**Table B1.** The Spearman's rank correlation coefficient between Mn-nodules/m$^2$, depth, and bathymetric derivatives

| Depth | Slope | TRI | Pl.C | Pr.C | TPI_B | TPI_M | TPI_F | Con. |
|---|---|---|---|---|---|---|---|---|
| 0.38 | 0.08 | 0.07 | 0.03 | 0.04 | 0.29 | 0.24 | 0.05 | -0.14 |

4.3.1 Effect of training sample size, ntree and mtry parameter: The descriptive statistics of the performance of each model were used as decision factors for the number of iterations. In all cases, the mean value with very low standard error, the very low standard deviation, range and the 95% confidence interval indicate a rather stable performance, without the need for further iterations.

**Table B2.** Descriptive statistics of MSR from different training set sizes, after 10 iterations with default settings.

| % Training Sample: | 10% | 20% | 30% | 40% | 50% | 60% | 70% | 80% | 90% |
|---|---|---|---|---|---|---|---|---|---|
| Mean | 34.8 | 30.2 | 26.1 | 23.3 | 22.2 | 21.3 | 19.7 | 18.3 | 18.1 |
| Std. Error | 0.1 | 0.0 | 0.0 | 0.0 | 0.0 | 0.0 | 0.0 | 0.0 | 0.0 |
| Median | 34.8 | 30.3 | 26.1 | 23.2 | 22.2 | 21.3 | 19.7 | 18.3 | 18.1 |





| Mode | 34.7 | 30.3 | 26.1 | 23.2 | 22.2 | 21.3 | 19.7 | 18.3 | 18.1 |
|---|---|---|---|---|---|---|---|---|---|
| Std. Deviation | 0.2 | 0.1 | 0.1 | 0.1 | 0.0 | 0.0 | 0.0 | 0.0 | 0.1 |
| Minimum | 34.5 | 30.1 | 25.9 | 23.2 | 22.1 | 21.2 | 19.6 | 18.2 | 18.1 |
| Maximum | 35.1 | 30.4 | 26.3 | 23.5 | 22.3 | 21.3 | 19.7 | 18.3 | 18.1 |
| C.I. (95.0%) | 0.1 | 0.1 | 0.1 | 0.1 | 0.1 | 0.0 | 0.0. | 0.0 | 0.0 |


**Table B3.** Descriptive Statistics of MSR from a different number of *ntree* parameter, after 10 iterations with 80% of the sample as training data and *mtry* = 3.

| *ntree:* | 100 | 200 | 300 | 400 | 500 | 600 | 700 | 800 | 900 | 1000 |
|---|---|---|---|---|---|---|---|---|---|---|
| Mean | 18.8 | 18.4 | 18.3 | 18.3 | 18.3 | 18.2 | 18.2 | 18.2 | 18.2 | 18.2 |
| Std. Error | 0.0 | 0.0 | 0.0 | 0.0 | 0.0 | 0.0 | 0.0 | 0.0 | 0.0 | 0.0 |
| Median | 18.8 | 18.4 | 18.3 | 18.3 | 18.3 | 18.2 | 18.2 | 18.2 | 18.2 | 18.2 |
| Mode | 18.8 | 18.4 | 18.3 | 18.3 | 18.3 | 18.2 | 18.2 | 18.2 | 18.2 | 18.2 |
| Std. Deviation | 0.1 | 0.1 | 0.1 | 0.1 | 0.0 | 0.1 | 0.1 | 0.1 | 0.0 | 0.0 |
| Minimum | 18.5 | 18.4 | 18.2 | 18.2 | 18.2 | 18.1 | 18.1 | 18.1 | 18.1 | 18.1 |
| Maximum | 18.9 | 18.5 | 18.5 | 18.4 | 18.3 | 18.3 | 18.3 | 18.3 | 18.2 | 18.2 |
| C.I. (95.0%) | 0.1 | 0.0 | 0.1 | 0.0 | 0.0 | 0.0 | 0.0 | 0.0 | 0.0 | 0.0 |

**Table B4.** Descriptive Statistics of MSR from different number of *mtry* parameter, after 10 iterations with 80% of the
sample as training data and *ntree* = 600.

| *mtry:* | 1 | 2 | 3 | 4 | 5 | 6 | 7 |
|---|---|---|---|---|---|---|---|
| Mean | 23.4 | 19.3 | 18.2 | 17.9 | 17.7 | 17.6 | 17.6 |
| Std. Error | 0.0 | 0.0 | 0.0 | 0.0 | 0.0 | 0.0 | 0.0 |
| Median | 23.4 | 19.3 | 18.2 | 17.9 | 17.7 | 17.6 | 17.6 |
| Mode | 23.4 | 19.3 | 18.2 | 17.9 | 17.7 | 17.6 | 17.6 |
| Std. Deviation | 0.0 | 0.1 | 0.1 | 0.1 | 0.0 | 0.0 | 0.0 |
| Minimum | 23.3 | 19.1 | 18.1 | 17.8 | 17.6 | 17.5 | 17.6 |
| Maximum | 23.5 | 19.4 | 18.3 | 17.9 | 17.7 | 17.7 | 17.7 |
| C.I. (95.0%) | 0.0 | 0.1 | 0.0 | 0.0 | 0.0 | 0.0 | 0.0 |





**Table B5.** Descriptive Statistics of MSR for the optimum selected RF model, after 30 iterations with 80% of the sample as training data, *ntree* = 600, and *mtry* = 6.

|  | Mean | Std. Error | Median | Mode | Std. Deviation | Minimum | Maximum | C.I. (95%) |
|---|---|---|---|---|---|---|---|---|
| **Optimum RF** | 17.6 | 0.0 | 17.6 | 17.6 | 0.0 | 17.5 | 17.7 | 0.0 |

**Table B6.** Descriptive statistics of validation measures for the optimum RF model, after 30 iterations with 80% of the sample as training data, *ntree* = 600, and *mtry* = 6 .

| RF Importance: | Depth | TPI_B | TPI_M | TRI | TPI_F | C | S | Pl.C | Pr.C |
|---|---|---|---|---|---|---|---|---|---|
| Mean | 80.1 | 63.6 | 46.7 | 36.1 | 24.5 | 19.7 | 12.0 | 2.6 | 2.4 |
| Std. Error | 0.1 | 0.1 | 0.1 | 0.0 | 0.0 | 0.0 | 0.0 | 0.0 | 0.0 |
| Median | 80.1 | 63.5 | 46.7 | 36.1 | 19.7 | 19.7 | 11.9 | 2.6 | 2.4 |
| Mode | 80.1 | 63.3 | 46.9 | 36.1 | 19.8 | 19.8 | 11.9 | 2.6 | 2.4 |
| Std. Deviation | 0.4 | 0.6 | 0.6 | 0.2 | 0.2 | 0.2 | 0.2 | 0.0 | 0.0 |
| Minimum | 79.1 | 62.6 | 45.0 | 35.7 | 19.2 | 19.2 | 11.7 | 2.5 | 2.3 |
| Maximum | 80.8 | 64.9 | 47.7 | 36.4 | 20.1 | 20.1 | 12.4 | 2.6 | 2.5 |
| C.I. (95.0%) | 0.1 | 0.2 | 0.2 | 0.1 | 0.1 | 0.1 | 0.1 | 0.0 | 0.0 |

### 4.3.2 Selection, application and external validation of the optimal model

Despite the fact that RF is a full non-parametric technique and there is no need for the residuals to follow specific assumptions (Breiman, 2001a), the examination of them can provide an in-depth look into its performance characteristics. The scatterplot of residuals against predicted values shows a random pattern, which is also confirmed by the low values of Pearson, Spearman, and $R^2$ coefficients between predicted values and residuals (Figure B1a and Table B6). Moreover, the residuals tend to cluster towards the middle of the plot without being systematically high or low, and having zero mean value (Figure B1 and Table B7). Their constant variance (homoscedasticity) implies that the distribution of error has the same range for almost all fitted values. Indeed, 99.3% of the residuals are inside the range ±15 and mainly the 81.2% inside the range ±5 (Table B8). The presence of outliers is very limited without affecting the main statistical characteristics of residuals (Table B7) indicating that the model adequately fits the overwhelming majority of the observations (>2165) and only random variation (that exists in any real natural phenomenon) or noise can occur.



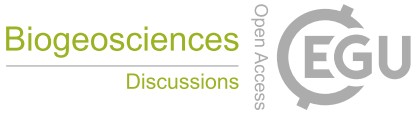

**Figure B1. S**catterplot between residuals and predicted values.

**Table B7.** Pearson, Spearman, and R² correlation coefficients between residuals and predicted values.

|  | Pearson | Spearman | R² |
|---|---|---|---|
| Correlation of residuals and predicted values | 0.1 | 0.2 | 0.0 |

**Table B8.** Main descriptive statistics of residuals and 5% trimmed residuals

|  | Mean | Std. Error | Median | Mode | Std. Deviation |
|---|---|---|---|---|---|
| Residuals | -0.2 | 0.1 | -0.2 | 0.6 | 4.4 |
| 5% Trimmed Residuals | -0.2 | 0.1 | -0.2 | 0.6 | 2.9 |

**Table B9.** Residuals range

| Residuals Range | ±20 | ±15 | ±10 | ±5 |
|---|---|---|---|---|
| % of Residuals | 99.8 | 99.3 | 96.1 | 81.2 |

4.3.4 RF importance

The production of the RF partial depedence plots, show the non-linear character between the Mn-nodules/m² and the

bathymetric derivatives.





**Figure B2.** Partial dependence plots for each of the predictor variables. The y axis represent the number of Mn-nodules/m²
and the x axis the values of each predictor variable (depth derivatives). The ticks inside the graphs indicate the deciles of the
data.

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
