# Peer review of "Quantitative mapping and predictive modelling of Mn-nodules' distribution from hydroacoustic and optical AUV data linked by Random Forests machine learning."

_Biogeosciences, 2018_

## Referee Comment (RC1) · Anonymous Referee #1 · 17 Aug 2018

General comments

This is a well written paper on the combined use of AUV imagery and acoustic surveys for the assessment of manganese nodules, which shows clear scientific and industrial relevance. However, it shows some similarities to Alevizos et al 2018 (similar approaches, but different locations). Both the size of the area covered and the number of images, highlight the use of AUVs and the importance of automated approaches for environmental assessment. The authors have made a point of being very transparent about their approach, and the statistical details provided in the appendices provide

extra confidence in the results presented (more of the statistical explanations of the results could be moved to the appendix, e.g. assessment of normality in 4.1).

Specific comments

(1) There is reference to a particular MBES depth data processing that guarantees removal of artifacts and improvements of georeferencing, but no reference or descriptions are given. Particularly in deep waters, inaccurate AUV positioning will be an issue, especially when trying to related photographs to 3m resolution bathymetric grids.

(2) As mentioned in line 420-425, choice of scale is important in deriving terrain metrics and a quantitative justification for the choice of chosen scales should be provided.

(3) The calibration of the model section is not as clear as it could be. Lines 223-225 need to clearly state that the default values were used for the assessment of training/testing sample size only. Lines 238 says that after training sample size was determined were mtry and ntree examined, but Line 241 mentions that for each case of different training sample size, ntree and mtry, the model was run ten times. The latter sentence should be split to clarify that each training sample size was not also tested for each different numbers of ntree and mtry. Similarly, Line 243, if I understood correctly, Appendix A only presents the averages for the 10 training size runs, and not the ntree and mtry runs. The wording here also needs to be clarified.

(4) I am not convinced that the approach taken can be used to determine the optimal training sample size proportion. More data is likely to yield better models, but by decreasing number of testing data points, one can also expect MSR to keep decreasing (as was shown here). A much more interesting question would be how many samples are needed to obtain accurate predictions.

(5) RF models also provide a measure of uncertainty, it would be interesting the provide uncertainty maps for the associated predictions and discuss potential spatial trends if any.

(6) Autocorrelation in Mn nodule distribution was discussed, but whether model residuals showed any spatial autoccorelation was nor assessed, nor were the effects of this autocorrelation on model assessment discussed.

(7) The discussion is very much focused on the model and although the exploratory nature of machine learning algorithm is mentioned, a little more discussion of the causation mechanisms (or potential hypothesis) would be valuable.

Technical corrections

Line 38. I would suggest changing sea bottom for seafloor

Lines 39-45 I would suggest specifically introducing the term backscatter, as I believe that to be one of the main data product used for to show Mn trends in regional surveys

Line 61 Reference style

Lines 81-83 Awkward, please rephrase

Line114 In the marine environment,

Line 131 remove scale)

Line 133 is deeper and has less relief

Line 189, while

Line 222 a threshold of 0.95 for correlation of variable seems very high

Line 256 in the study area,

Line 259 First sentence seems repetitive

Line 260 I do not think that the word 'alternation' here is the right one

Line 263 to 0.18%

Line 265 Awkward, please rephrase

[Figure]

Line 266 change approved by to supported

Line 270 measurements

Line 274 like Kriging

Line 275 area, and it is an important step

Line 276 and the produced bathymetric derivatives

Line 290 after a distance of 400m

Line 345 I would suggest to avoid finishing a sentence with too

Line 356 For our data,

Figure 12 b) for which mtry and c) for which ntree?

Line 385 Table 5 MAE, MSE and RMSE were not introduced previously, only MSR was mentioned in to method section

Line 413 The analysis of RF variable importance

Line 414 specific depth ranges

Line 415-417 I would suggest removing this sentence as it is not necessary

Line 418 All of them also contribute in a nonlinear way

Line 444 the study area, equal to

Lines 458-459 Awkward, please rephrase

Lines 468-471 Awkward, please rephrase

Line 471 Conversely,

Line 475 clues as to why

Line 491 Along these lines, several authors have included

Line 513 'a priori'

Line 516 as well as their size

Line 518 interest). Finally,

Line 561 the remaining derivatives

Lines 565-566 training and testing records

Line 570 Should be 4.1

Throughout, ground truth vs ground-truth, hydo-acoustic vs hydroacoustic, circa vs ca., space vs no-space between value and unit

---

## Referee Comment (RC2) · Anonymous Referee #2 · 24 Aug 2018

This paper was very interesting to read and clearly demonstrates how combining several state of the art scientific tools can achieve results that were, until recently, difficult to produce. The main idea in the manuscript; using Machine Learning to derive abundance of nodules from predictor variables remotely sensed with an AUV, has been applied by these authors and others datasets and this paper combines data and methodologies that have both been featured in other publications (cited in the paper). However, it presents a thorough protocol to make use of these tools, combine and optimize them, account for known caveats in the procedure and demonstrates the applicability of this

protocol in a practical situation. The scientific approach is complex but transparently detailed throughout the method, results and appendixes. Thus, this paper is a useful case study and a method that should be applicable to other similar datasets and, as such, is a valuable contribution to the exploration of the Manganese nodules fields in the CCZ. It is well written but could be streamlined and made easier to read. The important findings could be further highlighted in the results section by moving some of the subsections in the appendix (as highlighted by reviewer 1). In addition, I found that several sentences or groups of sentences in the discussion either were confused in their formulation or didn't make a clear point. Furthermore, the discussion could be structured into several paragraphs to help readers perceive the different points made by the authors.

I also have a couple of specific remarks and suggestion to add to those of reviewer 1:

R400: If RF is not good at predicting outside the ranges of the training set, could it affect the projected map of nodule abundance? Other studies projecting RF models (of species distribution) in space (or time) have used multivariate environmental similarity surfaces (MESS) maps (Elith et al. 2010). This procedure is mapping how dissimilar to known data points the predictors are across the projection area. This could potentially highlight that predictions in deeper and shallower areas than where nodule abundance samples are should be considered with care. This could also help target areas for future sampling. See Elith J, Kearney M, Phillips S (2010) The art of modelling range-shifting species. Methods in Ecology and Evolution 1:330-342

R415: The relevance of the depth as the most important predictor could be discussed further. Is there a geological reason why depth is the main driver of nodules distribution (as it looks unintuitive as to why such small changes in depth could drive nodule distribution)? Is it likely to be a proxy for another driver?

R499: Minor point but Judging by figure 12, the relation between MSR and the different tuning parameters, particularly the number of training samples is not linear and thus,

could either increase asymptotically towards a maximum or might continue increase logarithmically. Either way, It is unclear if more data would be a major improvement. Thus, collection of new data should focus on better distributed data

R510: Given the rarity of corers data compared to photo data, would it not be better to take all cores where there is photos to strengthen the comparison between the two nodule counting methods? The photos of areas where some of the cores have been taken can still be excluded from the RF model and externally validated afterward in order to make best use of available ship time and data.

And a few technical corrections and suggestions:

R56: "data points"? "Data sets"?

R180: could you specify what the correction would be?

R474: "resulting in biased results where Mn-nodules are bigger"?

R480: This is true, but is it necessary to state that here? Maybe it could be moved to the introduction

R476 - 485: It is hard to follow the authors point in that group of sentences. Do you mean that the observed influence of bathymetric factors on nodules distribution cannot necessarily be explained , but this observation is an interesting fact in itself it may later lead to further understanding of an underlying process?

R490: "as it ignores"?

R490: "To this end, several authors, have included the values of latitude/longitude and even LMI as predictor variables"?

516: "thus, high priority areas (e.g. these with highest commercial interest) can be targeted for sampling based on the results of optic data and RF modelling"?

Hope this is helpful

---

## Author Response (AR1)

Reviewer #1: General comments

This is a well-written paper on the combined use of AUV imagery and acoustic surveys for the assessment of manganese nodules, which shows clear scientific and industrial relevance. However, it shows some similarities to Alevizos et al 2018 (similar approaches, but different locations). Both the size of the area covered and the number of images, highlight the use of AUVs and the importance of automated approaches for

environmental assessment. The authors have made a point of being very transparent about their approach, and the statistical details provided in the appendices provide extra confidence in the results presented (more of the statistical explanations of the results could be moved to the appendix, e.g. assessment of normality in 4.1).

Authors comment:

We welcome all comments of Reviewer #1 and we appreciate the time and effort put to review this manuscript. Below we present our reply for each of the Reviewer's points:

We believe that the differences in this paper, compared with the paper from Alevizos et al, (2018), are not limited only to different locations. Alevizos et al, (2018) applied three different techniques in order to estimate the distribution of the Mn-nodules inside their study area. Two of them (Bayesian probability on beam backscatter and ISODATA classification) classify the bottom in areas with higher and lower number of Mn-nodules (based on backscatter values), while the third (RandomForests machine learning) predicts the Mn-nodule abundance in each location based on a number of predictor variables (MBES data) and training data (optic data). In our study, we focus only on the Random Forests machine learning prediction performance by applying and tuning the algorithm. Different predictor variables and in different scales (compared with Alevizos et al, 2018) were used; by doing so we supported the investigation of the role of predictor variables in different areas. It is worth to mention that in our study the absence of backscatter information as a predictor variable, showed that topographic factors alone can achieve relatively accurate predictions. Differently to Alevizos et al., an extensive statistical analysis (e.g. assessment of normality, spatial clustering) was performed in order to further investigate the distribution characteristics of the Mn-nodules. This analysis combined with the correlation analysis between the number of Mn-nodules/m2 and the derivatives highlighted the value of the Random Forests algorithm as a tool for complex spatial predictions. Furthermore, this study examined the distribution of the median size and its correlation with the number of Mn-nodules. The idea was to go one step further than Alevizos et al 2018, by introducing and applying a relatively simple

operational workflow, highlighting at the same time the advantages and disadvantages of the existing sampling procedures.

The assessment of normality in 4.1 was moved to Appendix B in order to strengthen the important results.  Finally, the paper by Alevizos et al.  was redrawn and will not be published in the way it has been discussed in Biogeoscience.    Reviewer #1: Specific comments

(1) There is reference to a particular MBES depth data processing that guarantees removal of artifacts and improvements of georeferencing, but no reference or descriptions are given. Particularly in deep waters, inaccurate AUV positioning will be an issue, especially when trying to related photographs to 3m resolution bathymetric grids.

Authors comment:

We added some more explanation in the text.

(2) As mentioned in line 420-425, choice of scale is important in deriving terrain metrics and a quantitative justification for the choice of chosen scales should be provided.

Authors comment:

Lines 420-423 state that the arbitrary choice of scale limits the value of the terrain metrics as explanatory variables exactly because of the scale dependency in environmental modelling.  Thus using derivatives of different scales (e.g.fine or broad scale BPI) contributions critically to environmental modelling results. Due to the lack of relevant literature for AUV scale data sets, the Concavity and Terrain Ruggedness indexes were created with the default scale of SAGA GIS v.6.3.0 (radius of 10 cells as stated in Table 1).  The three different values for the Topographic Position Index were selected based on the minimum possible correlation among them (surface correlation tool, in SAGA).

(3) The calibration of the model section is not as clear as it could be.  Lines 223-225 need to clearly state that the default values were used for the assessment of

C3

**BGD**

Interactive comment

Printer-friendly version

Discussion paper

training/testing sample size only. Lines 238 says that after training sample size was determined were mtry and ntree examined, but Line 241 mentions that for each case of different training sample size, ntree and mtry, the model was run ten times. The latter sentence should be split to clarify that each training sample size was not also tested for each different numbers of ntree and mtry. Similarly, Line 243, if I understood correctly, Appendix A only presents the averages for the 10 training size runs, and not the ntree and mtry runs. The wording here also needs to be clarified.

Authors comment:

Lines 223-225, 239-240 (in the submitted manuscript) were changed accordingly to the recommendations, stating clearly that the default RF values (for regression) were used only during the investigation of the optimum training size. Line 241 (in the submitted manuscript) was changed, now stating that ten different mtry and ntree values were applied for ten times each, only in the optimum selected training size. All tables regarding the statistical characteristics of the performance after 10 runs are presented in Appendix B.

(4) I am not convinced that the approach taken can be used to determine the optimal training sample size proportion. More data is likely to yield better models, but by decreasing number of testing data points, one can also expect MSR to keep decreasing (as was shown here). A much more interesting question would be how many samples are needed to obtain accurate predictions.

Authors comment:

Indeed, the less testing data points you have the more likely it is to achieve a lower error only because your model fits relatively well. In lines 356 – 361, we justify our choice to use the model with 80% because of the higher number of validation data compared to the 90% model.

(5) RF models also provide a measure of uncertainty, it would be interesting to provide

uncertainty maps for the associated predictions and discuss potential spatial trends if any.

Authors comment:

Indeed, the use of uncertainty prediction maps is a useful tool in spatial predictive mapping as they can reveal spatial trends (e.g. areas with higher uncertainty in predicted value) which might lead to additional sampling in order to advance the model and support the interpretation. The RF models inside the randomForests R package (Liaw and Wiener, 2002), can give an accurate prediction of the conditional mean of the response variable. The uncertainty (conditional quantiles) around this mean can be estimated by the use of the Quantile Regression Forests (Meinshausen, 2006), as they keep all the values in each node in each tree (not only the mean value), allowing the construction of prediction intervals. Quantile Regression Forests (QRF) models can be developed using the quantregForest R package (Meinshausen, 2012). The used MGET toolbox (Roberts et al, 2010) includes only the randomForests R package (Liaw and Wiener, 2002) and the party R package (Hothorn et al., 2006; Strobl et al., 2007; Strobl et al., 2008). MGET was selected as tool to keep the proposed workflow simple and, in a graphic environment familiar to many geoscientists. Recent comparative studies showed that the accuracy of the quantregForest R package against standard RF does not differ considerably, while it increased the computational time (Tung et al, 2014), without adding any other information regarding the variable importance. The use of other recently proposed methodologies as the Jackknife method (Wager et al, 2014), the Monte Carlo approach (Coulston et al, 2016) and U-statistics approach (Mentch and Hooker, 2016)) are far beyond of the aim and purposes of this paper.

1. Liaw, A. and Wiener, M.: Classification and regression by randomForest. R News, 2/3:18–22, 2002. http://CRAN.R-project.org/doc/Rnews/

2. Breiman, L.: Random forests. Machine Learning, 45, 5–32, 2001. https://doi.org/10.1023/A:101093340

3.Meinshausen, N.: 2 Quantile Regression Forests. Journal of Machine Learning Research 7, 983–999, 2006.

4.Hothorn T, Hornik K, Zeileis A.: "Unbiased Recursive Partitioning: A Conditional Inference Framework." Journal of Computational and Graphical Statistics, 15(3), 651–674, 2006. https://doi.org/10.1198/106186006X133933

5.Strobl, C. Boulesteix, A.L., Zeileis, A. and Hothorn, T.: Bias in random forest variable importance measures: Illustrations, sources, and a solution. BMC Bioinformatics, 8:25, 2007. https://doi.org/10.1186/1471-2105-8-25

6.Strobl, C., Boulesteix, A.L., Kneib, T., Augustin, T. and Zeileis, A.: Conditional variable importance for random forests. BMC Bioinformatics, 9:307, 2008. https://doi.org/10.1186/1471-2105-9-307

7.Tung N.T., Huang J.Z., Khan I., Li M.J., Williams G.: Extensions to Quantile Regression Forests for Very High-Dimensional Data. In: Tseng V.S., Ho T.B., Zhou ZH., Chen A.L.P., Kao HY. (eds) Advances in Knowledge Discovery and Data Mining. PAKDD 2014. Lecture Notes in Computer Science, vol 8444. Springer, Cham https://doi.org/10.1007/978-3-319-06605-9_21

8.Wager, S., Hastie, T., and Efron, B.: Confidence intervals for random forests: the jackknife and the infinitesimal jackknife. Journal of Machine Learning Research, 15(1):1625–1651, 2014.

9.Coulston, J. W., Blinn, C. E., Thomas, V. A., and Wynne, R. H.: Approximating prediction uncertainty for random forest regression models. Photogrammetric Engineering & Remote Sensing, 807 82(3):189 – 197, 2016.

10.Mentch, L. and Hooker, G. (2016). Quantifying uncertainty in random forests via confidence intervals and hypothesis tests. Journal of Machine Learning Research, 17(1):841–881.

(6) Autocorrelation in Mn nodule distribution was discussed, but whether model resid-

uals showed any spatial autocorrelation was nor assessed, nor were the effects of this autocorrelation on model assessment discussed.

Authors comment:

The spatial autocorrelation analysis of the residuals using the Global Moran's Index (same settings as Appendix A), showed low, but significant spatial autocorrelation (I=0.112112 p<0.01 and Z-score>2.58). The index number of the residuals is relatively low compared with the high initial values of the original data (I=0.69890 and I=0.697747 for the entire dataset and the 80% training dataset, respectively). The 5% trimmed residuals (see Appendix B-Table B8) showed that their spatial autocorrelation is only 0.093832. According to similar studies (i.e. regression RF), the presence of spatial autocorrelation in the residuals of the model can result in underestimation of the true prediction error (Ruß und Kruse, 2010). The presence of low spatial autocorrelation values in the residuals of regression RF has been reported also by other authors (e.g. Mascaro et al, 2014; Xu et al, 2016); it is a common problem in all the well-established machine learning methods (e.g. RandomForests, Neural Network, Gradient Boosting Machine, and Support Vector Machines) when dealing with regression predictions of spatial variables (Gilardi and Bengio, 2009; Ruß und Kruse, 2010; Santibanez et al, 2015 a,b). The spatial plotting and visual examination of the residuals (Figure 1) showed that this spatial clustering exists mainly in the small sub-area b, and especially in the areas which are associated with an increased slope (>3°), where the AUV is forced to vary its altitude between the ascending and descending phase (Figure 7b) and consequently affects the image quality and the later modelling results.

1.Ruß, G., and Kruse, R.: Regression Models for Spatial Data: An Example from Precision Agriculture. CDM 2010. Lecture Notes in Computer Science, vol 6171. Springer, Berlin, Heidelberg, 2010. https://doi.org/10.1007/978-3-642-14400-4_35

2.Mascaro, J., Asner, GP., Knapp, DE., Kennedy-Bowdoin, T., Martin, RE., Anderson, C., Higgins, M., and Chadwick, D.: A Tale of Two "Forests": Random Forest Ma-

chine Learning Aids Tropical Forest Carbon Mapping. PLoS ONE 9(1): e85993, 2014. https://doi.org/10.1371/journal.pone.0085993

3.Xu, L., Saatchi, SS., Yang, Y., Yu, Y., and White, L.: Performance of non‑parametric algorithms for spatial mapping of tropical forest structure. Carbon Balance Manage, 11:18, 2016. https://doi.org/10.1186/s13021-016-0062-9

4.Gilardi, N., and Bengio, S.: Comparison of four machine learning algorithms for spatial data analysis. Conf. Signals Syst. Comput., 17, 160–167, 2009.

5.Santibanez, S., Lakes, T., and Kloft, M.,: Performance Analysis of Some Machine Learning Algorithms for Regression Under Varying Spatial Autocorrelation. The 18th AGILE International Conference on Geographic Information Science, Lisboa (Portugal), 9-12 June, 2015a.

6.Santibanez, Sebastian F., Marius Kloft and Tobia Lakes. "Performance Analysis of Machine Learning Algorithms for Regression of Spatial Variables. A Case Study in the Real Estate Industry." the 13th International Conference of GeoComputation, Dallas (USA), May 20 – 23, 2015b.

(7) The discussion is very much focused on the model and although the exploratory nature of machine learning algorithm is mentioned, a little more discussion of the causation mechanisms (or potential hypothesis) would be valuable.

Authors comment:

Classic studies have shown that the bathymetry and the variation of the topographic characteristics of the seafloor affects the sediment deposition environment, bottom currents and thus also geochemical processes in the sediment. All these factors determine Mn-nodule growth and thus affect the distribution of Mn-nodules on regional scales (e.g. Craig, 1979; Sharma and Kodagali, 1993; ). It is unknown how these properties influence the Mn-nodule distribution on meter to tens of meters scales as seen in our AUV data. The non-linear relationship between Mn-nodules and bathymetry on

such high-resolution scales only started very recently (Peukert et al, 2018 and also th withdrawn submission by Alevizos et al). To elaborate more on the hydrodynamic and geochemical reasons behind the observed distribution pattern, we would need more investigations at and in the sediment on the same scale. Without such data, any elaboration on the reasons for the distribution would be purely speculative, without additional 'ideas' than the known and published influencing parameters. 1.Craig, J. D.: The relationship between bathymetry and ferromanganese deposits in the north equatorial Pacific, Marine Geology, 29, 165–186, 1979. https://doi.org/10.1016/0025-3227(79)90107-5

2.Sharma, R. and Kodagali, V.: Influence of seabed topography on the distribution of manganese nodules and associated features in the Central Indian Basin: A study based on photographic observations, Marine Geology, 110, 153–162, 1993. https://doi.org/10.1016/0025-3227(93)90111-8 3.Peukert, A., Schoening, T., Alevizos, E., Köser, K., Kwasnitschka, T., and Greinert, J.: Understanding Mn-nodule distribution and evaluation of related deep-sea mining impacts using AUV-based hydroacoustic and optical data. Biogeosciences, 15, 2525-2549, 2018. https://doi.org/10.1007/978-3-319-57852-1_24

4.Alevizos et al, Schoening T., Koeser K., Snellen M. and Greinert J.: Quantification of the fine-scale distribution 1 of Mn-nodules: insights from AUV multi-beam and optical imagery data fusion. Biogeosciences Discussions. pp. 1-29, 2018. https://doi.org/10.5194/bg-2018-60

Reviewer #1: Technical corrections - Authors comments:

Line 38 I would suggest changing sea bottom for seafloor - Done

Lines 39-45 I would suggest specifically introducing the term backscatter, as I believe that to be one of the main data product used for to show Mn trends in regional surveys - Done

[Figure]

Line 61 Reference style - Done

Lines 81-83 Awkward, please rephrase - Done

Line114 In the marine environment, - Done

Line 131 remove scale) -Done

Line 133 is deeper and has less relief - Done

Line 189, while - Done

Line 222 a threshold of 0.95 for correlation of variable seems very high – Done (we changed the term highly with perfectly correlated. In similar studies even higher thresholds have been used during the selection of predictor variables (Che Hasan et al, 2014; Li et al, 2016; Li et al, 2017)).

1.Che Hasan, R., Ierodiaconou, D., Laurenson, L., Schimel, A.: Integrating Multibeam Backscatter Angular Response, Mosaic and Bathymetry Data for Benthic Habitat Mapping. PLoS ONE 9 (5), e97339, 2014. https://doi.org/10.1371/journal.pone.0097339

2.Li J, Tran, M, Siwabessy, J: Selecting Optimal Random Forest Predictive Models: A Case Study on Predicting the Spatial Distribution of Seabed Hardness. PLoS ONE 11 (2): e0149089, 2016. https://doi.org/10.1371/journal.pone.0149089

3.Li, J., Alvarez, B., Siwabessy, J., Tran, M., Huang, Z., Przeslawski, L., Radke, L., Howard, F. and Nichol, S.: Application of random forest, generalised linear model and their hybrid methods with geostatistical techniques to count data: Predicting sponge species richness. Environmental Modelling & Software, 97, 112-129, 2017. https://doi.org/10.1016/j.envsoft.2017.07.016

Line 256 in the study area, - Done

Line 259 First sentence seems repetitive - Done

Line 260 I do not think that the word 'alternation' here is the right one – Done (now:

[Figure]

change)

Line 263 to 0.18% - Done

Line 265 Awkward, please rephrase - Done

Line 266 change approved by to supported - Done

Line 270 measurements - Done

Line 274 like Kriging - Done

Line 275 area, and it is an important step - Done

Line 276 and the produced bathymetric derivatives - Done

Line 290 after a distance of 400m - Done

Line 345 I would suggest to avoid finishing a sentence with too - Done

Line 356 For our data, - Done

Figure 12 b) for which mtry and c) for which ntree? - Done

Line 385 Table 5 MAE, MSE and RMSE were not introduced previously, only MSR was mentioned in to method section - Done

Line 413 The analysis of RF variable importance - Done

Line 414 specific depth ranges - Done

Line 415-417 I would suggest removing this sentence as it is not necessary - Done (removed from here and added to the discussion part as we consider that is important to refer that other authors have found such relationships in the marine and terrestrial environment).

Line 418 All of them also contribute in a nonlinear way - Done

Line 444 the study area, equal to - Done

[Figure]

Lines 458-459 Awkward, please rephrase - Done

Lines 468-471 Awkward, please rephrase - Done

Line 471 Conversely, - Done

Line 475 clues as to why - Done

Line 491 Along these lines, several authors have included - Done

Line 513 'a priori'- Done

Line 516 as well as their size - Done

Line 518 interest). Finally, - Done

Line 561 the remaining derivatives - Done

Lines 565-566 training and testing records - Done

Line 570 Should be 4.1 - Done Throughout, ground truth vs ground-truth, hydro-acoustic vs hydroacoustic, circa vs ca., space vs no-space between value and unit: - Done (ground-truth, hydroacoustic, ca. and space between value and unit, were selected)

Please also note the supplement to this comment:
https://www.biogeosciences-discuss.net/bg-2018-353/bg-2018-353-AC1-supplement.pdf

[Figure]

[Figure]

**Figure 1.** Spatial plotting of the RF residuals (absolute values). The intervals of their range are in accordance with the Table B9 (Appendix B) in the submitted manuscript.

**Fig. 1.**

[Figure]

Biogeosciences Discuss.,
https://doi.org/10.5194/bg-2018-353-AC2, 2018

[Figure]

*Interactive comment on* "Quantitative mapping
and predictive modelling of Mn-nodules'
distribution from hydroacoustic and optical AUV
data linked by Random Forests machine learning"
*by* Iason-Zois Gazis et al.

**Iason-Zois Gazis et al.**

igazis@geomar.de

Reviewer #2 General comments:

This paper was very interesting to read and clearly demonstrates how combining sev-
eral state of the art scientific tools can achieve results that were, until recently, difficult
to produce. The main idea in the manuscript; using Machine Learning to derive abun-
dance of nodules from predictor variables remotely sensed with an AUV, has been ap-
plied by these authors and others datasets and this paper combines data and method-

ologies that have both been featured in other publications (cited in the paper). However, it presents a thorough protocol to make use of these tools, combine and optimize them, account for known caveats in the procedure and demonstrates the applicability of this protocol in a practical situation. The scientific approach is complex but transparently detailed throughout the method, results and appendixes. Thus, this paper is a useful case study and a method that should be applicable to other similar datasets and, as such, is a valuable contribution to the exploration of the Manganese nodules fields in the CCZ. It is well written but could be streamlined and made easier to read. The important findings could be further highlighted in the results section by moving some of the subsections in the appendix (as highlighted by reviewer 1). In addition, I found that several sentences or groups of sentences in the discussion either were confused in their formulation or didn't make a clear point. Furthermore, the discussion could be structured into several paragraphs to help readers perceive the different points made by the authors.

Authors comments:

We welcome all comments of Reviewer #2 and we appreciate the time and effort put to review this manuscript. Below we present our reply for each of the reviewer's points:

Similar to Reviewer #1, Reviewer #2 highlights the transparent, thorough and well-written workflow, and notices also that this methodology has been applied in the past, as well as the need for a different structure of some parts in the manuscript. Both comments have been already answered to Reviewer #1 and are considered in the revised version. In addition, we followed the recommendation of the Reviewer #2 to divide the discussion part into several paragraphs in an effort to state clearly the points of our study.

Reviewer #2 Specific comments:

I also have a couple of specific remarks and suggestion to add to those of reviewer 1: R400: If RF is not good at predicting outside the ranges of the training set, could it

affect the projected map of nodule abundance? Other studies projecting RF models (of species distribution) in space (or time) have used multivariate environmental similarity surfaces (MESS) maps (Elith et al. 2010). This procedure is mapping how dissimilar to known data points the predictors are across the projection area. This could potentially highlight that predictions in deeper and shallower areas than where nodule abundance samples are should be considered with care. This could also help target areas for future sampling. See Elith J, Kearney M, Phillips S (2010) The art of modelling range-shifting species. Methods in Ecology and Evolution 1:330-342

Authors comments:

Doubtlessly, the 'weakness' of the RF method is to predict outside of the range of the training set, this can influence the accuracy of the final abundance map. The need for extrapolation is always given in deep ocean studies by the limited numbers of actual samples. The problem of having not 'entirely' representative samples can only be solved by collecting a great number of sample points (like images in our case) that are well-distributed inside the study area (i.e. data that will include the entire range of the number of Mn-nodules/m2 and they are come from all the different sub-terrains). The comparative use of different machine learning algorithms (Support Vector Machines and Artificial Neural Networks) for the same dataset, which are able to extrapolate beyond the training range (e.g. Balabin and Lomakina, 2011; Martious and Lambert, 2017), can reveal the size of this 'weakness' in RF predictions. Such extrapolated predictions should be treated carefully regarding their accuracy and should always been validated with samples from the outer parts (lower and upper) of the training range. The main difficulty of our approach, is the need for different representative large training data in every different study area. The use of multivariate environmental similarity surfaces (MESS) can contribute to Mn-nodule exploration, by indicating other similar Mn-nodules fields in the wider area, based on the similarity of morphological characteristics of the already studied areas. To our knowledge the combined use of RF and MESS has not been applied yet as. Elith et al. (2010) used Boosted Regression Tree

(BRT) and Maximum Entropy (MaxEnt) machine learnings approaches; an approach interesting for future studies. Another promising, although complex would be the use of the Transfer Learning Approach. This approach can overcome the drawback of traditional machine learning, in which the training predictive algorithms should be trained each time based on previously collected (labelled or unlabeled) data from the study area. By using Transfer Learning, one can take an already trained model and transfer the part of the model that contains the necessary built relationships into a new model (usually smaller) that has to learn only the extra relationships/patterns that may exist in the new study area (e.g. Pan and Yang, 2010; Lu et al, 2015). Thus, the non-linear relationship between the number of Mn-Nodules/m2 and the topographic factors can be transferred and applied to other potential areas, where there is a lack of labelled optic data, and may include slightly different bathymetric range and topographic characteristics.

1. Balabin, R.M. and Romakina, E.I.: Support vector machine regression (LS-SVM) - an alternative to artificial neural networks (ANNs) for the analysis of quantum chemistry data? Phys. Chem. Chem. Phys., 13, 11710–11718, 2011. https://doi.org/10.1039/c1cp00051a

2. Martius, G., Lampert, C.H.: Extrapolation and learning equations. CoRR abs/1610.02995, 2016. http://arxiv.org/abs/1610.02995

3. Elith, J., Kearney, M. and Phillips, S.: The art of modelling range-shifting species. Methods in Ecology and Evolution, 1, 330–342, 2010. https://doi.org/10.1111/j.2041-210X.2010.00036.x

4. Pan, S. J., and Yang, Q.: A Survey on Transfer Learning. IEEE Transactions on knowledge and data engineering, Vol.22, No. 10, 1345-1359, 2010.

5. Lu, J., Behbood, V., Hao, P., Zuo, H., Xue, S., Zhang, G.: Transfer learning using computational intelligence: A survey. Knowledge-Based Systems, 80, 14–23, 2015. http://dx.doi.org/10.1016/j.knosys.2015.01.010

[Figure]

R415: The relevance of the depth as the most important predictor could be discussed further. Is there a geological reason why depth is the main driver of nodules distribution (as it looks unintuitive as to why such small changes in depth could drive nodule distribution)? Is it likely to be a proxy for another driver?

Authors comments:

This question is also asked in the specific comment (7) from Reviewer #1, and it is answered there.

R499: Minor point but Judging by figure 12, the relation between MSR and the different tuning parameters, particularly the number of training samples is not linear and thus, could either increase asymptotically towards a maximum or might continue increase logarithmically. Either way, It is unclear if more data would be a major improvement. Thus, collection of new data should focus on better-distributed data

Authors comments:

Indeed, it is not clear if more data would be a major improvement. The availability of more data and especially if they were better distributed, would most likely reinforce the model to build better and wider relationships between the predictor and response variables. This would allow keeping a larger number of validation data points. The need for more and better-distributed data has been stated in lines 498 – 504, especially when considering the spatial clustering inside the study area. The influence of the number of training data for model performance still remains a discussion point between studies showing an improvement by adding more data (e.g. Bishop, 2006), and other studies presenting stable performance of the model even if more data are added (e.g. Zhu et al, 2012).

1. Bishop, C.M.: Pattern Recognition and Machine Learning. Information Science and Statistics. Springer, Heidelberg, 2006.

2. Zhu, X., Vondrick, C., Ramanan, D., and Fowlkes, C. C.: Do We Need More Training

[Figure]

Data or Better Models for Object Detection? In BMVC, 3, 5, 2012.

R510: Given the rarity of corers data compared to photo data, would it not be better to take all cores where there is photos to strengthen the comparison between the two nodule counting methods? The photos of areas where some of the cores have been taken can still be excluded from the RF model and externally validated afterwards in order to make the best use of available ship time and data.

Authors comments:

In an ideal scenario with a sufficient number of available box-corers (many box corers, in which we are referring in the manuscript), both scenarios should be applied. The greatest number of them should be deployed in areas with photos in order to calculate better the factor between counted Mn-nodules in photos and in box-corers, and the rest in areas without photos in order to estimate the accuracy of the model in areas far away from the optic data but still inside the study area. However, in a realistic scenario the amount box-core samples will always be limited and thus they should be deployed in areas with photos to establish a better relationship between these two quantitative methods.

And a few technical corrections and suggestions -Authors comments:

R56: "data points"? "Data sets"? – Done we use data sets

R180: could you specify what the correction would be? - Done. This correction can be a simple factor that describes the ratio between the number of Mn-nodules seen in the photo and the number of nodules counted in box-corers (considering for the different spatial scales). Kuhn and Rathke (2014) used this approach, but also considered two different nodule size spectra.

1. Kuhn, T., Rathke, M.: Report on visual data acquisition in the field and interpretation for SMnN. Deliverable D1.31 of the EU-Project Blue Mining. BGR Hannover, 34 pp. 2017. www.bluemining.eu/downloads

[Figure]

R474: "resulting in biased results where Mn-nodules are bigger"? – Done (This phrase is in lines 472-473). The sentence has been changed.

R480: This is true, but is it necessary to state that here? Maybe it could be moved to the introduction

R476 - 485: It is hard to follow the authors point here. Do you mean that the observed influence of bathymetric factors on the nodule distribution cannot necessarily be explained? This observation is an interesting fact in itself and may lead to a better understanding of an underlying Mn-nodule formation process?

R490: "as it ignores"? – Done

R490: "To this end, several authors, have included the values of latitude/longitude and even LMI as predictor variables"? – Done (based on Referee's #1 suggestion)

516: "thus, high priority areas (e.g. these with highest commercial interest) can be targeted for sampling based on the results of optic data and RF modelling"? – Done

Please also note the supplement to this comment:
https://www.biogeosciences-discuss.net/bg-2018-353/bg-2018-353-AC2-supplement.pdf
* * *
[Figure]

[revised manuscript text omitted]